# Multi-variable flood damage modelling with limited data using supervised learning approaches

Dennis Wagenaar[1], Jurjen de Jong[1], Laurens M. Bouwer[1]

[1] Deltares, Delft, The Netherlands

*Correspondence to*: Dennis Wagenaar (dennis.wagenaar@deltares.nl)

**Abstract.** Flood damage assessment is usually done with damage curves only dependent on the water depth. Several recent studies have shown that supervised learning techniques applied to a multi-variable dataset can produce significantly better flood damage estimates. However, creating and applying a multi-variable flood damage model requires an extensive dataset, which is rarely available and this is currently holding back the widespread application of these techniques. In this paper we enrich a dataset of residential building and content damages from the Meuse flood of 1993 in the Netherlands, to make it suitable for multi-variable flood damage assessment. Results from 2D flood simulations are used to add information on flow velocity, flood duration and the return period to the dataset, and cadastre data is used to add information on building characteristics. Next, several statistical approaches are used to create multi-variable flood damage models, including regression trees, bagging regression trees, random forest, and a Bayesian network. Validation on data points from a test set shows that the enriched dataset in combination with the supervised learning techniques delivers a 20% reduction in the mean absolute error, compared to a simple model only based on the water depth, despite several limitations of the enriched dataset. We find that with our dataset, the tree-based methods perform better than the Bayesian Network.

## 1 Introduction

Decision making in flood risk management is increasingly based on studies that quantify the flood risk rather than only the flood hazard. Flood damage estimation is therefore becoming increasingly important (Merz et al, 2010).Flood risk assessment supports policy makers to decide which flood risk management measures are most efficient in reducing flood risks and how much investment is cost-efficient. With the European Union Floods Directive (EC, 2007) now fully in place, national flood risk assessments are being developed with the final aim to support flood risk management plans. In the Netherlands, such flood damage assessment has been used to derive the optimal protection standard for flood protection (Kind, 2013; van der Most, 2014), using the current Dutch standard method for damage modelling (Kok et al., 2005). Also for insurance applications, more precise estimates of flood damages are required.

Flood risk assessments require flood damage models. These models typically predict the damage as fraction of the potential damage, based on the water depth, and average building repair and replacement costs for different types of buildings (Messner et al., 2007; Jonkman et al., 2008). Similar approaches are also applied to other natural hazards, for example for landslides (Papathome-Köhle et al., 2015) and the software package HAZUS can be used for floods, earthquakes and

hurricanes (Scawthorn et al., 2006). Alternative approaches to calculate flood risk do also exist, such as vulnerability indicators (Papathoma-Köhle, 2016).

Simple flood damage models often don't perform well, as shown by their validation (e.g. Jongman et al., 2012). This is because water depth alone cannot explain the full complexity of the flood damaging processes and several studies have only found low correlation coefficients (typically below 0.5) between the water depth and the flood damage (e.g. Merz et al., 2013, Pistrika and Jonkman, 2009). Furthermore, often no local data is available on flood damage and therefore a relationship between the water depth and damage either needs to be estimated or transferred from other areas (Wagenaar et al., 2016). This can cause errors as simple models hold many implicit assumptions that may not be valid for the situation the model is transferred to. For instance, Elmer et al. (2010) showed that an event with a low flood probability could not use the same damage function as a flood event with a high probability. These implicit assumptions cause large unexplained differences between flood damage functions (Wagenaar et al., 2016; Gerl et al., 2016). Transferability however could be improved, when a model describes more variations of the damaging process, and when more variables are included in the damage models (e.g. flood probability is explicitly part of the model). Similar problems are also present in the modelling of other natural hazards. For example Fuchs et al. (2007) found that building materials are very important for debris flow damage modelling and that models can therefore not always be transferred in space and time.

Current approaches suffer from two main limitations: first, they rely on limited information and usually only take into account water depth as a predictor, and use a deterministic relation between water depth and some fraction of average maximum damages; secondly, they are deterministic in nature, while it has been shown that uncertainties in this approach are large, but generally not quantified e.g. in the Dutch standard method (Egorova et al., 2008). Some of the multi-variable methods are able to provide probability distributions, rather than deterministic estimates of damages.

Recently, multi-variable flood damage models have been created with a German dataset based on telephone interviews. Thieken et al.(2005) found that apart from the water depth also the contamination of the flood water and precautionary measures were important to estimate the flood damage. In Thieken et al. (2008) these extra variables were included in a simple multi-variable flood damage model as a surcharge. Using information from this same database, Merz et al. (2013) used regression and bagging trees and Vogel et al. (2014) used Bayesian Networks to predict the flood damage. Spekkers et al. (2014) applied regression trees to estimate pluvial flood damage. Van Oostegem et al.(2015) applied the Tobit estimation technique to a multi-dimensional dataset in Belgium to estimate pluvial flood damages. These multi-variable flood damage models have been shown to perform better than simple flood damage models in Schröter et al. (2014) (up to 25% reduction in mean absolute error, MAE), both tested on their own dataset and on datasets from other floods (Schröter et al., 2014). Also, some multi-variable approaches (Bayesian Networks, Bagging trees and Random Forests) generate probability distributions of estimated damages, and thus provide information on uncertainties of the estimates. Therefore, multi-variable flood damage models look like a promising approach to improve flood damage modelling.

The application of multi-variable flood damage models for flood risk management studies is still difficult because of the large data requirements. Running a multi-variable flood damage model for a new area requires for every object several

variables on the flood hazard and building characteristics that are not yet typically collected. Also creating new multi-variable flood damage models is currently rarely done because they also require records of flood damages at building level.

More commonly available (although still rare) are simple datasets that hold records with the flood damage that occurred for each building with sometimes a few other variables (such as location or water depth). Such datasets may have been created

for compensation purposes or to build simple flood damage models but may miss most of the desired variables. An example of such a dataset is the flood damage dataset collected after the Meuse flood of 1993 in the Netherlands which is used here. Previously this dataset has been described in Wind et al. (1999) and in more detail in WL Delft (1994).In this paper we will explore the use of supervised learning techniques to build flood damage models based on a dataset that is very different from the datasets used in previous studies (i.e. the German dataset applied by Merz et al. (2013) and Schröter et al. (2014)).) The

dataset in this paper was collected by insurance experts directly after the flood for compensation purposes and covers all affected buildings. This is different from the German data which was collected a year after the flood for research purposes based on a sample of the affected buildings. The data is also different in that in the original study only a few variables were collected, in contrast for the German dataset all variables (except return period) were based on telephone interview answers. In this study several methods are applied to enrich the Meuse 1993 flood damage dataset with extra flood hazard and

building characteristic variables. We will answer the question of whether this enriched dataset from a different source then previous studies is also suitable to build a multi-variable flood damage model. The expectation is that the multi-variable models perform better than a model based on a single variable (water depth) and that even data with limited quality will improve the results.

2D hydraulic simulations of the 1993 flood on the Meuse are used to enrich the dataset with additional flood characteristics.

Cadastre data is used to enrich the Meuse dataset with extra building characteristics. Four different supervised learning techniques are then applied to this enriched dataset: a regression tree, bagging regression trees, random forest and a Bayesian network. A part of the dataset will be held back and will only be used for validation. This validation is then used to determine whether the enriched dataset combined with supervised learning techniques performs better than a traditional damage function based on the original dataset of water depths. In this paper we will focus on predicting absolute flood

damages rather than relative flood damages. This is because the exact building values are not available.

## 2 Methods and data

### 2.1 Datasets

#### 2.1.1 Meuse 1993 damage dataset

The dataset available for this research is based on the Meuse flood of 22 December 1993 in the Province of Limburg in the Netherlands (WL Delft, 1994). Although no dike breaches occurred in this event, several towns and urban areas located

close to the river were affected. The flood caused a total of 254 million guilder (price level 1993) in direct damages, which is approximately 180 million euros today (price level 2016). The flood inundated 180 km$^2$, which is about 8% of the Province of Limburg. 32% of the damage pertains to residential buildings and content (furnishings). In this study only residential damage is considered. Other major damage categories were business (29%), government (24%) and agriculture (8%) (WL
Delft, 1994). These categories are not considered because they are more heterogeneous and less data about them is available. Damage information was collected in the context of a compensation arrangement for flood damages by the national government. All data was collected by sending damage experts from insurance companies to the affected buildings, several weeks after the flood event had occurred. Directly after the damage data was collected in 1994, the data was shared with WL Delft (now Deltares) to create a flood damage model. WL Delft received 5780 records for damage to residential buildings.
The damage to privately owned residential buildings was collected by an organisation called "Stichting Watersnood 1993" , the damage to companies and the structure of rental residential buildings was collected by another organisation called "Stichting Watersnood Bedrijven 1993". So, in this set up of the damage collection, the building structure of rental residential buildings was collected by "Stichting Watersnood bedrijven", the organization that collected company damages. This is different from the organization that collected the rest of the residential damages. The structure damage to rental
residential buildings was only shared with WL Delft (1994) in some partial aggregate form. WL Delft (1994) presumably distributed this partially aggregated rental residential building damage over the individual rental residential buildings. The exact method for this was however not reported and the original dataset is no longer available. Therefore, we had to work with a dataset which includes unknown manual actions. The structure damage data is therefore from inconsistent quality, the content damage however has no such problems. Furthermore, it is expected that the percentage of rental residential buildings
in the affected area of Limburg is relatively low, limiting the impact of this data problem.

Another issue with the dataset is that for privacy reasons the exact locations of the buildings were not shared with WL Delft. Only the 6 digit postal code was available for this study, which makes it difficult to enrich the dataset, as between 1 and 20 buildings share the same 6 digit postal codes in the dataset.

In the original dataset the water depth (relative to the ground floor level) was estimated by the experts that surveyed the
damage. The quality of the water depth estimate is questioned by WL Delft (1994; report 9, appendix A) because it was not the main aim of the survey and the experts visited several weeks after the water had receded. A plot of the water depth (see Fig. 1) and the damage doesn't show an obvious relation. The correlation between the water depth and the damage is weak (Pearson correlation coefficient = 0.18).

The final dataset also contains information on the number of inhabitants per building, whether the house has a basement and
whether the house was attached to other houses. However, this data is not described in any of the available reports so the collection methods are not known, but the recorded values are clear enough to incorporate in this study. Two more variables are also included in the WL Delft dataset and also not described in any available report. These are emergency actions and ownership of the house. The meaning of the values found in the dataset for these variables is however not sufficiently clear, and could unfortunately not be taken into account in this study.

### 2.1.2 Upgraded Meuse 1993 dataset

To improve the dataset, additional information is required on both the flood hazard and exposure variables. The results of a 2D flood simulation and cadastre data were used to upgrade the dataset, in terms of hazard and exposure information, respectively. Because no observational data is available on flood characteristics other than the water depth, a simulation of the flood event was done. In the 2D flood simulation tool WAQUA (Rijkswaterstaat, 2013), a verified model of the state of the Meuse during the 1993 flood was available (Becker, 2012) and applied in this study to get extra variables. Using this model, a new simulation was run using a discharge boundary condition at Eijsden and a water level boundary condition at Keizersveer for the period 1 November 1993 to 31 Januari 1994. This simulation was used to create a maximum water depth map, a flood duration map, a flood return period, and a flow velocity map at a spatial resolution varying between 10 and 40 meters.

The maximum water depth and flow velocity are standard outputs of WAQUA. Flood duration is however not a standard output and is more difficult to get from a 2D flood simulation because the drainage also needs to be included in the schematisation (Wagenaar, 2012). During the 1993 Meuse flood, most drainage occurred because of the natural slope in terrain and therefore the 2D flood simulation implicitly includes most of the drainage because the discretised bed level is included. The flood duration can then be calculated by analysing the time-varying maps of the water depth and calculating for every cell the time between the moment a cell is inundated and the moment the cell is dry again. However, some cells in the digital elevation map in WAQUA are surrounded by cells that have a higher elevation. These cells do not drain in the 2D flood simulation and are still inundated at the end of the simulation. For these cells the flood duration has been calculated based on the change in water depth. If the water depth in a cell stays the same in the simulation for 24 subsequent hours the cell is considered dry at the moment this stable water depth is first reached.

Simulations were also ran with the same Meuse 1993 schematisation for design discharges with 1, 10, 50, 100, 250 and 1250 return periods. These discharges are based on HR2006 (Diermanse, 2004) and have discharges of respectively 1300, 2260, 2869, 3109, 3431 and 4000 $m^3$/s. The results of these simulations were combined to create a flood return period map for the Meuse 1993 situation. This map shows for each cell at what return period it first floods. Figure 2 shows that large water depths occurred and that most of the area floods frequently. The majority of the houses is however located in the safest areas with the lowest water depths and highest return periods.

These maps (water depth, flow velocity, flood duration and return periods) were linked to the original damage records using cadastre data. The data of the cadastre has exact building locations, postal codes, living area within the residential buildings, the building footprint area and the construction year. The building year was used to filter the data to find the building stock of 1993. Then, based on the building locations the 2D flood simulation results were linked to the cadastre data.

This combination of cadastre data and 2D flood simulation data is then used to make the link with the original flood damage records. First per postal code a list is made of the damage records in the postal code area and ranked based on the water depth in the original damage records. Then another list is made of the objects per postal code according to the cadastre and

also ranked based on the simulated water depth. The cadastre objects combined with the 2D flood simulation data is then linked per postal code based on the water depth rank. This results in a join between the original damage records, cadastre data and 2D flood simulation results. Table 1 gives an overview of the available records in this combined dataset.

The method of joining cadastre objects with damage records within a postal code area based on water depth rank is error prone. The modelled water depth is on average 30 cm larger than then the recorded water depth. This is possibly because the difference in reference level of both data sources as the recorded water depth is relative to the floor level and the modelled water depth is relative to the digital elevation map. Not all houses have the same floor elevation and both the recorded and the modelled water depth are uncertain, because of recording and model imprecisions. It is therefore likely that some damage records have been linked to the wrong object. However, errors will likely be limited, because the join on postal codes is accurate. Object and flood variables are generally similar for buildings within the same postal code area (e.g. houses within a street are typically similar to each other) so these errors are expected not to significantly disturb the general trends in the data. The errors are therefore considered acceptable given that the purpose of the dataset is only to build a flood damage model. If significant errors are present this would result in a reduced performance of the supervised learning algorithms on the test set. A relatively simple alternative to this water depth rank method is also applied. In this alternative, the average value at all building locations in the postal code area was assigned to each of the objects in the postal code.

## 2.2 Supervised learning algorithms

Several supervised learning techniques have been applied to the enriched dataset to build multi-variable flood damage models. The different supervised learning techniques all have different ways to generalize the training data in such a way that it can give useful predictions of the total damage.

These multi-variable flood damage models are be compared to two different reference models to assess the value of the enriched dataset and to assess the value of multi-variable flood damage models in general. Below the different supervised learning algorithms applied are described in further detail.

### 2.2.1 Regression: Root function

The first reference model only uses the square root of the water depth (see formula 1) to predict the flood damage. This model represents the damage functions commonly applied today in flood risk management studies because many damage functions have approximately the shape of a root function (e.g. Scawthorn, C., et al., 2006; Thieken et al., 2008; Penning-Rowsell et al., 2005; Sluijs et al., 2000). Merz et al. (2012) applied the same method to get a reference damage function. The purpose of this reference model is to see the benefits of using more data.

The root function (1) is fitted to the dataset in such a way that the coefficients $c_1$ and $c_2$ are optimised to get the smallest possible error based on the total damage (td) and water depth (wdf) data. The values of the coefficients are optimized for the best fit with the ordinary least squares method. This is done with the Python package SciPy.

$$td = c_1 + c_2\sqrt{wdf} \qquad \textbf{(1)}$$

### 2.2.2 Multi-variable linear regression

The second reference model uses multi-variable linear regression to fit a linear model to the data. This model represents more simple/traditional techniques to make a multi-variable model from data. The purpose of this reference model is to see the benefits of potentially better techniques to build multi-variable models from data. Multi-variable linear regression is for example used in Islam (1997) to make multi-variable flood damage models. Linear regression is used without transformations of the input variables, because there is no clear indication that in the data that there are non-linear relationships (for example see figure 1).

To ensure that the model captures general trends and doesn't fit too strongly to the observed data (overfitting) the LASSO technique is used. This technique determines the coefficients in such a way that a penalty is applied for increasing the coefficients and using the variables more. LASSO yields sparse models, so some coefficients will become zero, which means they are not useful for the prediction. Therefore, the LASSO technique is useful for variable selection. To make this work correctly the data is normalized before training the model.

The multi-variable linear regression was carried out with the Scikit learn library in Python (Pedregosa et al. 2011). LASSO requires an alpha parameter to be set which determines the height of the penalty applied. Several alpha values were tried (0, 0.5, 1and10). The model is very insensitive to the Alpha value (all formulations perform about equally well), an alpha value of zero performs best on all indicators. Therefore, it is not optimized further and the alpha is set to zero. When alpha is zero the method is equal to the ordinary least square method and no overfitting prevention is in place and LASSO is not necessary. This shows that overfitting is not an issue for relatively simple techniques such as linear regression with this dataset and number of variables.

### 2.2.3 Regression tree learning

Decision trees are a way to represent complex relationships between data and classes in a tree structure. A decision tree can be seen as a series of binary questions (nodes) leading to an answer in the form of a class (leaf). A question can be related to any variable at any value (e.g. is the water depth smaller than 0.5m).
A regression tree is similar to decision trees but instead of classes it results in real numbers. In theory, regression trees can be very large and have a separate leaf for each unique value in the dataset. However, it is more common to combine several similar unique values inside the same leaf and represent it with a summary statistic number (mean). In such a case the regression tree is an approximation of the relationship.

Regression tree learning algorithms can create optimal regression trees based on a dataset. In this paper the dataset consists of 4398 flood damage records (incomplete records are discarded) with 11 variables for each damage record (see table 1). The regression tree algorithm aims to split the dataset into subsets in such a way that the mean squared error (MSE) of the predicted total damage for all observations is minimized compared to the observed data. It does this by calculating the

reduction for all candidate splitting variables according to their value and then picking the combination that maximises the MSE reduction ($\Delta I$), this is shown in (2). $n$ is the total number of observations in the node, $\mathbf{y}_n$ is the vector of observed target

values in the node and $\bar{y}$ is the mean of the target values in the node. $\mathbf{y}_{nL}$ and $\mathbf{y}_{nR}$ are vectors with the observed target values of the left and right group after the split and $\bar{y}_L$ and $\bar{y}_R$ are the mean observed target value for the left and the right group. The

regression tree is grown by repeating this process at each node of the tree. This has been done with the Scikit learn library in Python (Pedregosa et al. 2011).

$$\Delta I = \frac{1}{n}\left(\sum (y_n - \bar{y})^2 - \sum (y_{nL} - \bar{y}_L)^2 - \sum (y_{nR} - \bar{y}_R)^2\right) \qquad (2)$$

A regression tree algorithm keeps splitting the dataset into new branches until no more reductions in the MSE can be made. This can result in overfitting, which results in very large trees with only one data point per leaf. These very large trees are not a realistic representation of reality, and they typically perform badly when they have to predict the damage for a new data

point that wasn't used for building the tree. There are several methods to prevent overfitting. The simplest methods require a minimum number of data points in a leaf or set a maximum number of nodes that the tree is allowed to contain. The disadvantage of these methods is that they sometimes don't build out a branch within the tree which at first doesn't look promising but which can make valuable homogeneity improvements deeper in the tree. A method called pruning is a more sophisticated method, in which the entire tree is first build with a subset of the data points, and then cut back based on its

performance on data points that were not used for building the tree. The tree is cut back by removing the nodes by their performance improvement (least performing nodes first), the optimal pruning depth is than picked by testing the different pruning depths on the test set. This method was investigated in this research. This was done using *Matlab's 'Statistics and Machine Learning Toolbox'* (Matlab website), based on the work by Breiman et al. (1984), because the Python libraries do not support pruning. The MAE was applied as metric to find the optimal pruning depth. The performance of the pruning

algorithm on this dataset was similar to a regression tree built with a combination of a minimum data point requirement per leaf and a maximum number of leaves (MAE with pruning in Matlab is 0.55 against 0.56 without pruning in Python). Therefore, the rest of the study was performed without pruning in the Scikit learn library in Python (Pedregosa *et al.2011*). Accordingly, the results shown do not include pruning.

### 2.2.3 Bagging regression trees

Another method to avoid overfitting and generally improve the accuracy of decision/regression trees is bootstrap aggregating, also called bagging. The idea behind the method is to resample the dataset multiple times and to build a new

regression tree for each resampled dataset. This results in an ensemble of regression trees. The resulting flood damage is then the average of the ensemble of regression trees. Resampling is done by building several datasets by randomly picking records from the original dataset (each record is allowed to be used multiple times in the same dataset). Every resampled dataset therefore randomly leaves out a fraction of the observations and puts more weight on other observations because they are picked multiple times. Bagging regression trees also lead to probabilistic outcomes because the ensemble of trees can be seen as a probability distribution of the outcome.

### 2.2.4 Random Forest

A random forest is a more advanced variation of bagging regression trees. Apart from building multiple trees with resampled datasets it also randomly excludes a subset of variables at each decision split. This will result in an ensemble of regression trees each based on a different set of damage records and each leaving out a different number of variables at each decision split. For this paper the default settings of Scikit learn are applied, in our case this means 8 variables are left out at each decision split.

### 2.2.4 Bayesian Network

A Bayesian Network is a type of Probabilistic Graphical Model that represents a set of random variables and their conditional dependencies in a directed acyclic graph (DAG) structure. Each variable in the network may be observed or represented as a prior probability distribution and dependencies between variables are represented with edges representing joint probability distributions. The edges in a Bayesian Network are directed which means there is a direction in which the influence of one variable flows to the other. From this network, inference can be done in order to use knowledge of one variable to make predictions about other variables.

Bayesian Networks and Probabilistic Graphical Models in general are used in many different fields, such as bioinformatics (e.g. Mourad et al. (2011), image processing (e.g. Sudderth & Freeman, 2008) and speech recognition (e.g. Bilmes, 2002). Recently, they have also been applied to flood damage modelling (Vogel et al., 2014; Schröter et al. 2014; Van Verseveld, 2014). Schröter et al. (2014) found that their performance is often better than that of the different types of tree methods. Furthermore, a Bayesian Network can give its result as a probability distribution and does not require information about each variable in order make predictions. If fewer variables are available, the Bayesian Network handles this by adjusting the probability distribution of the outcome. This makes it ideal for transfer of models to other locations where less data is available than for the location where the model was originally based on. Furthermore, it returns (for each object) probability distributions rather than deterministic values, which is valuable for assessing uncertainties within the damage model estimates.

A Bayesian Network can be discrete, continuous or a combination. In this paper fully discrete Bayesian Networks are used, in which all variables are discretized into bins. Given a network the probability of particular set of discrete variable values can be calculated with the following formula:

$$P(X_i, \ldots, X_n) = \prod_{i=1}^{n} P(X_i | parents(X_i)) \tag{3}$$

Where $X_i$ are the variables and $parents(X_i)$ is the set of variables directed to $X_i$. The probability of a single variable value
can be obtained by taking the sum of all the probabilities that contain the variable value of interest. The conditional probabilities are stored in conditional probability tables (CPTs). These tables show, for each combination of parent variable values, the probability of each possible output value.

A data-driven Bayesian Network can derive all its CPTs from the data and even derive its graph structure from the data. For this paper, two Bayesian Networks were made: A data-driven Bayesian Network with both the graph structure and the CPTs
derived from the dataset and an expert network where the graph structure was estimated in an expert session but the CPTs were derived from the dataset. All calculations were done with a Python library called libpgm (Cabot, 2012). This library follows the methodology described in Koller and Friedman (2009).

The CPTs are learned with maximum likelihood estimation. This method estimates the (joint) probability distributions based on the number of observations. The discretisation assumptions have an impact on the maximum likelihood estimation. If the
variables are discretised into a large number of bins more possible combinations of states are possible. These combinations of states grow exponentially with the number of bins of the parent variables. A too fine discretisation therefore quickly leads to more possible states than available data points. This results in a poor performance of the maximum likelihood estimation. Koller and Friedman (2009) call this one of the key limiting factors in learning Bayesian Networks from data. A too coarse discretisation on the other hand is also not desirable because it limits the precision of the Bayesian Network. For this study a
balance was found by trying several discretisation resolutions until the best result was found based on the MAE criterion.

Discretisation was done by splitting the data into bins with an equal number of data points in each bin. This works better than making equal sized bins because of the large extremes in especially the damage data. Equal sized bins would either increase the number of bins, which is detrimental to the maximum likelihood estimation (having bins that contain no observations), or the bins would be so large that a majority of the data points would end up in the same bin, which would
limit the Bayesian Network performance. The number of bins per variable was chosen based on the performance of a test set on the MAE criterion. This was done by varying the discretisation of the most important variables until the smallest error was found. For the Bayesian Network with the data-driven structure the number of bins chosen was slightly larger, because the network is less complex than the expert network.

The performance of the Bayesian Network on the testing data can be sensitive for discretisation. There are two possible
alternatives for the discretisation method applied in this paper: An optimisation algorithm could be applied to determine the optimal discretisation, or a continuous Bayesian Network could be used (Friedman and Goldszmidt, 1996). Apart from solving the discretization problem the advantage of a continuous Bayesian Network is that it would probably perform better

in predicting extreme values but a disadvantage is that the Bayesian Network is restricted to specific families of parametric probability distributions (Friedman and Goldszmidt, 1996). An optimization algorithm for the discretization can minimize the error produced by the discretizing but does not solve the fundamental problem of having too few data points.

The data-driven structure is also learned with the libpgm Python library. This library is using a constrained-based approach for structure learning, as is described in Koller and Friedman (2009). In a constrained based approach the structure is learned by calculating dependencies and conditional dependencies among the variables. When two variables are dependent regardless of what they are conditioned by, an edge (connection) is formed. The algorithm follows this procedure to create the entire network. The result is shown in figure 4 (left).

As an alternative to the data-driven structure a structure was also made in an expert meeting involving several Deltares flood damage/Bayesian Network experts (see acknowledgements). In the expert meeting the network was constructed based on a combination of expert judgement/logic and with the knowledge of figure 3 in this paper. The experts focused mainly on edges that they thought are relevant for estimating the flood damage. The result is shown in Figure 4 (right).

The relationship between the total, structural and content damage is known and not probabilistic: total damage = structure damage + content damage. Also, in our case the structure damage, content damage and total damage are always all dependent variables. Therefore, using a Bayesian Network to model this exact definitional relationship could only introduce extra errors and not add anything extra explanation.Therefore in the expert network it has been decided not to use the total damage variable. Instead the total damage is calculated as the sum of the expected value of the structure and the content damage. In the data-driven network the structure damage was not included by the algorithm. Therefore, the total damage variable itself is used for the data-driven network.

The advantage of an expert based network is that experts focus on the connections that matter most rather than on all possible connections. Furthermore, experts can include connections that are not found in this dataset but are expected to exist in theory or in an independent test set. The advantage of a learned network is that new and previously unknown relationships between variables can be discovered.It is expected that the Bayesian Networks in this manuscript are not very sensitive to overfitting during the CPT learning. Koller (2008) only mentions overfitting in the maximum likelihood estimation of Bayesian Networks in relation to discretization that is too fine and offers no techniques to counter overfitting in the maximum likelihood estimation. This expectation that overfitting isn't an issue was tested by testing the Bayesian Network on its own training data. If overfitting is an issue the model should do much better in predicting its own data then in predicting new data. This isn't the case (for the expert model) the MAE is even slightly worse when calculated on its own data (0.622), the correlation coefficient and $R^2$ are only slightly better (0.24 and 0.04) and only the mean bias error (MBE) is significantly better (-0.015). See results section for comparison.

## 2.3 Variable importance

In order to investigate the value of more data it is interesting to study the contribution of the different variables to the prediction accuracy. This can be done with bagging trees and the random forests methods. This importance can be calculated as the (normalized) total reduction of the mean square error brought by the different variables as achieved during the training of the models. This can be used to compare the relative importance of the variables among each other. This feature importance can be calculated for all the regression trees in the ensemble and a general importance is computed by the sci-kit learn library by taking the average of the feature importances in the tree. This was applied in this study for the bagging trees. The variable importance has been separated for predicting the importance of the total damage, structural damage and the content damage. For the calculation of the variable importance the dataset is used in which the average per postal code is used for the new variables. The water depth rank is not used because it could transfer some of the importance of the original water depth value to the new variables.

Another way to study variable importance is with the LASSO technique in multi-variable linear regression. LASSO can drop unimportant variable coefficients to zero. If a variable is dropped to zero it means the variable is less important.

## 3 Results

### 3.1 Model comparison

The different models are tested on a test set that was not used for training the models. Four indicators are used to rate the performance of the models: Mean Absolute Error (MAE), Mean Bias Error (MBE), the Pearson correlation coefficient, and the coefficient of determination ($R^2$). The MAE is the mean absolute error divided by the average damage, so a smaller MAE is a better model. The MBE is the average error, this differs from the MAE in that an overestimation is able to correct for an underestimation and the other way around. A low MBE shows that the sum of a large number of predictions will probably be very accurate. The Pearson correlation coefficient is a measure of the linear dependence between two variables. This measure is used to compare the predicted damages with the actual damages in the test set. A Pearson correlation of one means a perfect correlation, zero means no correlation and minus one a perfect inverse correlation. $R^2$ is the predictive capacity of a model compared to just using the average damage as a prediction. If the R2 is zero it means the independent variables add no predictive capacity compared to just using the average. When R2 is 1 it means the independent variables can explain all variation in the dependent variable. Table 3 shows the results for the different models.

Table 3 shows that given that the models can use all data, random forest and bagging regression trees perform best and equally well. These two methods reduce the MAE by 12% compared to a reference model using the same data (multi-variable linear regression). Bagging regression trees and Random Forest do perform significantly better than normal regression trees, as was also noted by Merz et al. (2013) for flood damages in Germany. Random Forest and Bagging regression trees also outperform the Bayesian Networks. The normal regression tree also works better than the Bayesian

Networks. This contradicts earlier findings by Schröter et al. (2014), who found that in most cases Bayesian Networks outperformed the regression trees. Schröter et al. (2014) did however have a very different dataset from the one applied in this study.

Many explanations are possible for the relatively poor performance of the Bayesian Networks. The discretization of the data is a possible problem. Some trends could be too subtle to be captured by the rough discretization, but not enough data points are available for a more precise discretization. Perhaps there still is some space for improving the discretization, for example by applying an optimization algorithm to pick bin definitions in such a way that the available information is applied optimally (Vogel et al. 2012 applied such an algorithm). Another possible reason is that Bayesian Networks might be more sensitive to low quality data in combination with a small dataset. Some of the CPTs applied in the Bayesian Networks here are large and conditional probabilities are based on a relatively small number of observations. Some wrong observations may then have a relatively large impact on the damage prediction.

In the data-driven network the variable of interest (total damage) in our test is only influenced by the water depth. This is because the water depth relative to the ground floor is known while the content damage is not known, so the known water depth blocks all the influence of other variables and the unknown content damage has no influence because it is unknown (it is a target variable). The data-driven Bayesian Network is therefore in our test in practice only dependent on the water depth. So the structure learning decides to ignore the other variables when the water depth relative to the ground floor is available. This is probably because the data-driven structure algorithms finds all variables equally important and therefore draws only the most important edges (connections) regarding the total damage. Other methods (e.g. as described by Riggelsen, 2008) for structure learning might be able to give better results.

The multi-variable linear regression reference model does a good job on the MBE but is clearly weaker on the other performance indicators, which shows that for predicting aggregate damages for e.g. policy studies, the more complex methods are less beneficial. This is different in cases where individual building damages are important, for instance for insurance rating purposes. The reference root function has a very large bias compared to the other models. This is probably because the shape of the root function is inappropriate for this flood event.

**3.2 Benefits of more data**

The models were trained with different numbers of variables to see whether the additional data is valuable. As expected, the best performing model with a high number of variables always performs significantly better than the best performing model with fewer variables. More data therefore seems to add potential value to the damage prediction despite the possible quality issues in the additional data. The MAE of the best performing model with only the water depth (regression tree) can be reduced by a further 14% by the best model using all data (Random Forest). The MAE of the root function fitted to the data (representing common practice) can be reduced by about 20% using the Random Forest with all data.

The method to join the extra data with the original data based on water depth rank is not effective. Just taking the average value per postal code appears to work better. The water depth rank probably sometimes assigns extreme variable values to the wrong objects which disturb some correlations in the data.

### 3.3 Variable importance

The total importance of variables that were added in this study is about 30% (figure 5), that means that 30% of the error reduction during the training of bagging tree model originates from variables that were added to the dataset. The added variables therefore clearly help to improve the prediction accuracy. This assessment was done without the water depth ranking join because this could assign some of the importance of the original water depth to the modelled water depth. The original water depth is by far the most important variable. Construction year is an important variable for the structure

damage but not for the content damage. This is as expected. Household size is quite important for the structural damage but insignificant for the content damage. This is less obvious but it could be that large families live on average in larger houses but do not have much more valuable contents on the ground floor. Return period is an important variable for both the structure and the content damage. This was also expected because the population in areas that flood more frequently are expected to have more flood experience, thus resulting in better preparedness and lower damages. This effect is visible in the

data, with return period having an importance of about 10%.

For the best fitting multi-variable linear regression model (LASSO alpha=0) no variables are dropped. Only when the alpha is increased to 10, 5 variables are dropped, however this also causes a slight drop in model performance (MAE goes from 0.578 to 0.588). The dropped variables are: Building footprint, building age, living area, flood duration and flow velocity. From these dropped variables, two have a significant importance in the bagging tree variable importance assessment. These

are building age and living area. It could be that those variables are more important in non-linear models.

### 4 Discussion and conclusion

Additional data improves flood damage modelling relative to a test set, even if this data comes from a collection of different sources and is of limited quality (error prone). The supervised learning algorithm is also important. Given the same data

there are large differences between the algorithms. Random Forests and bagging regression trees perform significantly better than normal regression trees and multi-variable linear regression. The Bayesian Networks perform poorly compared to any of the tree based methods.

Our current approach doesn't show that the additional variables are beneficial for the Bayesian Networks. However, because the tree methods can benefit from the additional data it is likely that in some cases Bayesian Networks could also. The poor

performance of the Bayesian Networks contradicts earlier studies (Schröter et al., 2014) and could be due to the discretization method, quality of the expert network, network learning algorithm or problems with data quantity or quality.

The test set that was applied in this paper for the validation of the model, was randomly selected from the data and consistently applied among all models. The accuracy of the indicators for model performance could perhaps be further improved through some form of cross-validation. Also the tweaking of different models could become more accurate if cross-validation was used instead of validation on a single test set only. For example, the optimization of the stop criteria for

tree based models and the alpha value in the LASSO method for the multi-variable linear regression could be improved that way.  Expectations are that this would cause minor improvements in results but that it would not influence the conclusions of this paper.

This paper did not address another benefit of Bayesian Networks, Random Forest and Bagging trees, which is the incorporation of uncertainty. Bayesian Networks do this explicitly in the method and for Bagging Trees or Random Forest

each tree can be seen as a possible damage estimate and together the trees represent a probability distribution.

The methods applied in this manuscript provide an uncertainty estimate for a single object. For policy decision making it is often useful to aggregate these uncertainty estimates to a total uncertainty for the entire flood event. This can be done with the assumption that all objects are perfectly correlated to each other (one tree will apply to the entire event but what tree is uncertain), or with the assumption that all objects are independent of each other (each object will have a different tree but

what tree is uncertain). Both assumptions are however not completely correct (Wagenaar et al., 2016). The Bayesian Network framework might offer a middle way to model this correctly. If each object has a copy of the original Bayesian Network, and these Bayesian Networks are linked together based on the location of the objects, it can be explicitly taken into account that nearby objects are more likely to have similar damages. This could be an argument to prefer Bayesian Networks over tree based methods in the future.

The dataset applied in this paper had many limitations. The most important limitation is that the exact locations of the objects are unknown. Because of this, it was difficult to link building and flood characteristics to damage records. An attempt to do this by using water depth rank performed worse than just using the average variable values per postal code. Despite this limitation, the added data still produced significantly better damage estimates. Another problem with the dataset is the unknown manual adjustment to an unknown share of data (rental residential buildings) for the structural damage

records. These actions may have introduced a relationship between structural damage and some of the originally recorded variables that wasn't there in reality. This could in theory cause a slight overestimation in the prediction performance of the models on the test set. This effect on the results is however expected to be small, because most of the prediction improvements came from adding variables that were not available for doing the manual actions in 1994.

This study applied absolute damages rather than relative damages. This requires the supervised learning algorithms to

implicitly also predict information about the values at risk besides the vulnerability. The algorithms can do this with variables such as living area, footprint area, building year and household size. This seems less error prone and better than estimating such values at risk with general rules of thumb based on assumptions about construction costs and content value. Such assumptions could cause extra errors, and therefore in this study absolute damages were used.

This paper trained flood damage models on just a single flood event. Ideally training data should consist of multiple events so that the spectrum of possible damages which the model is trained upon is larger. Especially for the transfer to other areas this would be important. Models that are trained on a single event could overfit on this event and this problem would not show up if the model is tested with data from that same event (even if this specific data wasn't used for training the model).

A good example of this appears in the good performance of the regression tree based on only the water depth versus the fitted root function based on only the water depth. The root shape of a damage function which many expert models use (see section 2.2.1) and which makes physically sense, is performing much worse than a more flexible model that can adjust to other relationships between damage and water depth. This is explained by figure 6 which shows a downward sloping damage function after 90cm of water depth, a shape very different from damage functions normally found in the literature. The root

function model therefore starts producing large errors after 90 cm while the regression tree can capture this trend well. This downward sloping makes physically no sense but could be explained by other variables such as return period. Return period could be a proxy for flood experience and better preparation because houses that experienced large flood depths in 1993 are probably on lower ground and also experience floods in general more often. This relationship is probably not true for other types of events, for example large flood depths due to dike breaches. So in that sense, the regression tree is overfitting on

this single flood event.

Supervised learning can help to create and improve flood damage models. They have many theoretical advantages over deterministic damage functions based on only the water depth. The application of supervised learning in flood damage modelling remains challanging in practice, because of limited data availability. In this paper we utilized different data sources compared to previous studies to acquire this data and showed that also on this dataset the methods are beneficial,

especially the tree based methods. Future work may merge available datasets from different events and from different countries in order to develop a model that can be applied using several hazard variables, and which also works in circumstances outside areas for which flood damage data is available.

**Acknowledgments**

We thank our colleagues Kathryn Roscoe for advice on the Bayesian Networks and our colleagues Karin de Bruijn and Marcel van der Doef for their input in constructing the expert Bayesian Network. We would also like to thank the editor (Dr. Thaler) and the reviewers (one anonymous reviewer and Dr. Fuchs), their detailed constructive comments and suggestions helped to substantially improve this paper. This research has received funding from the European Union's Horizon 2020 research and innovation programme under Grant Agreement number 641811 (Improving predictions and management of

hydrological extremes – IMPREX), see also http://www.imprex.eu.

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

**Table 1: Description of the variables in the flood damage dataset for the Meuse flood of 1993.**

|  | Variable | Unit | Source | Pearson correlation on total damage |
|---|---|---|---|---|
| td | Total damage | Guilder (1993 value) | Original dataset[a] | 1 |
| sd | Structure damage | Guilder (1993 value) | Original dataset[a] | 0.85 |
| cd | Content damage | Guilder (1993 value) | Original dataset[a] | 0.83 |
| df | Water depth relative to floor | m | Original dataset[a] | 0.18 |
| dg | Water depth relative to DEM | m | Flood simulation[b] | 0.18 |
| bs | Basement | 1=Yes, 2=No | Original dataset[a] | -0.04 |
| dh | Detached house | 1=Yes, 2=No | Original dataset[a] | 0.08 |
| hs | Household size | Number | Original dataset[a] | 0.17 |
| fv | Flow velocity | m s$^{-1}$ | Flood simulation[b] | 0.04 |
| fd | Flood duration | h | Flood simulation[b] | 0.05 |
| rp | Return period | year | Flood simulation[b] | -0.09 |
| ba | Building age | year | Cadastre[c] | 0.01 |
| la | Floor area for living | m$^2$ | Cadastre[c] | 0.04 |
| fa | Footprint area building | m$^2$ | Cadastre[c] | -0.02 |

[a] WL Delft, 1994

[b] 2D flood simulation data using WAQUA

5  [c] Basisregistraties Adressen en Gebouwen (BAG), version 2011 (Kadaster website).

**Table 2: Results of different models for four indicators: MAE, MBE, $R^2$ and correlation coefficient. The models had access to all variables (except for the root function). The version of the dataset with the water depth rank join between the old and the new variables is used .**

| Calculation | MAE | MBE | $R^2$ | Correlation coefficient |
|---|---|---|---|---|
| Root function | 0.612 | 0.194 | 0 | 0.15 |
| Multi-variable linear regression | 0.578 | 0.055 | 0.07 | 0.27 |
| Regression tree | 0.561 | 0.065 | 0.03 | 0.31 |
| Bagging regression tree | 0.504 | 0.061 | 0.15 | 0.38 |
| Random forest | 0.508 | 0.054 | 0.16 | 0.39 |
| Data-driven Bayesian Network | 0.629 | 0.525 | 0 | 0.21 |
| Expert Bayesian Network | 0.607 | -0.08 | 0.03 | 0.21 |

**Table 3: The best performing model based on the MAE indicator with different number of variables.**

| Variables | Method | MAE | MBE | $R^2$ | Correlation coefficient |
|---|---|---|---|---|---|
| Only water depth | Regression tree | 0.564 | 0.071 | 0.08 | 0.306 |
| Only original variables (waterdepth, household size, detached house, basement) | Bagging trees | 0.551 | 0.052 | 0.07 | 0.345 |
| All variables (water depth rank join) | Random Forest | 0.508 | 0.054 | 0.16 | 0.394 |
| All variables (average postal code join) | Random Forest | 0.488 | 0.035 | 0.17 | 0.41 |

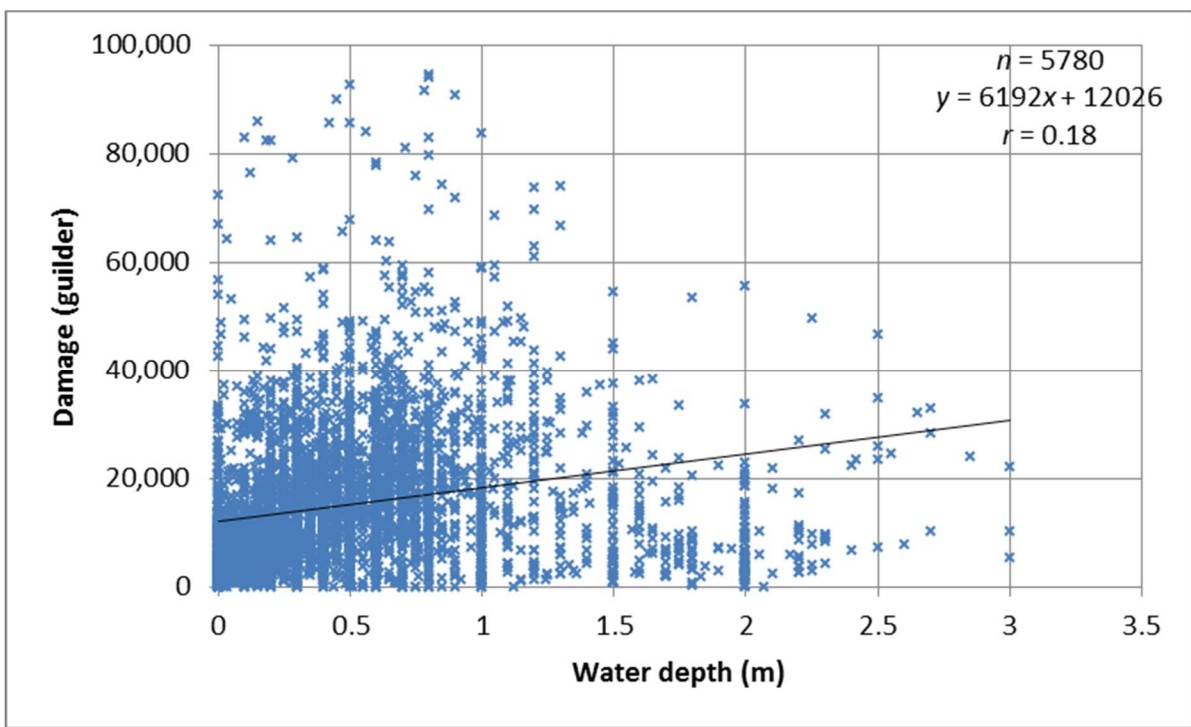

**Figure 1: Scatter plot showing the relation between water depth and damage in the original data set..**

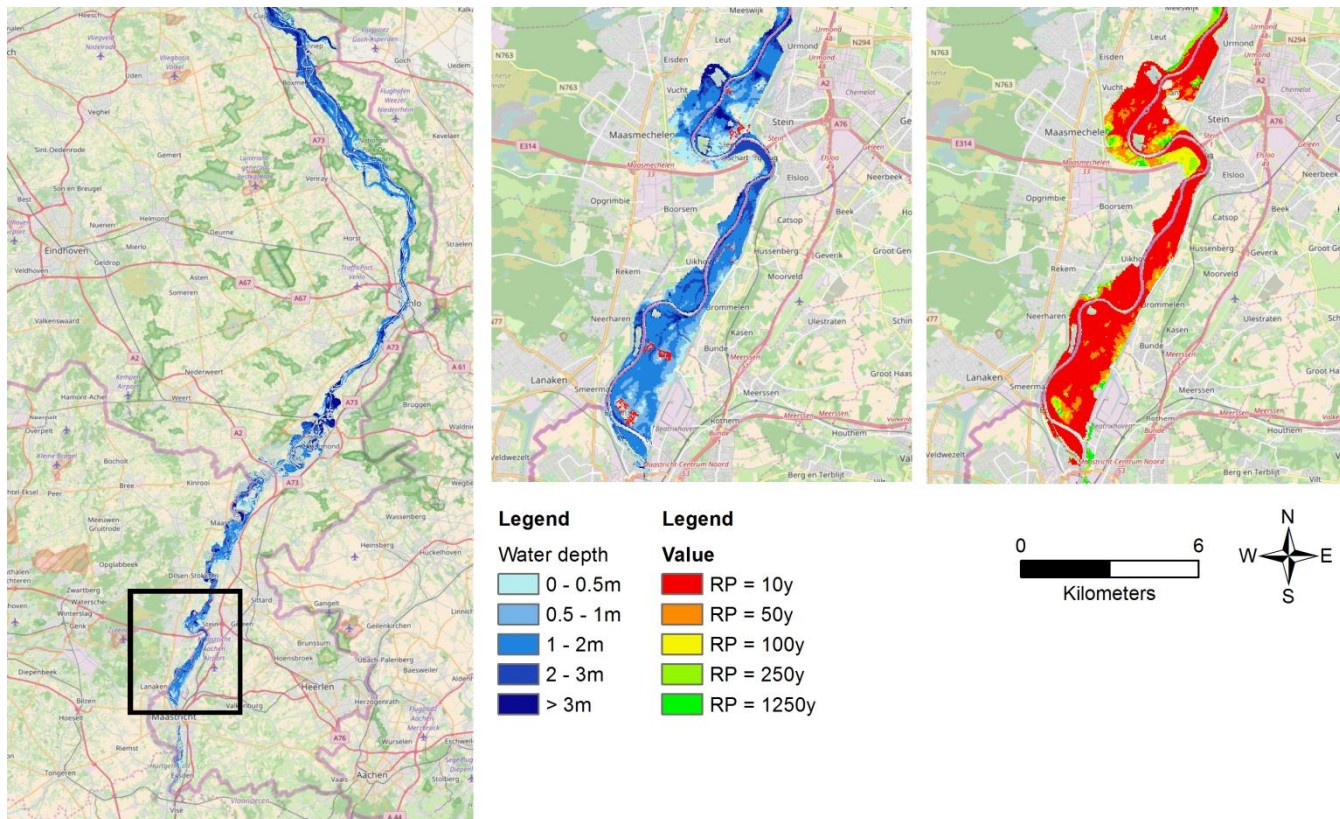

**Figure 2: Left the simulated water depth for the entire study area in Limburg. In the center the simulated water depth and affected population (in red) for an example area. On the right the return period at which areas start flooding for the example area. The example area is defined in the box in the left picture. The scale bar corresponds to the example area.**

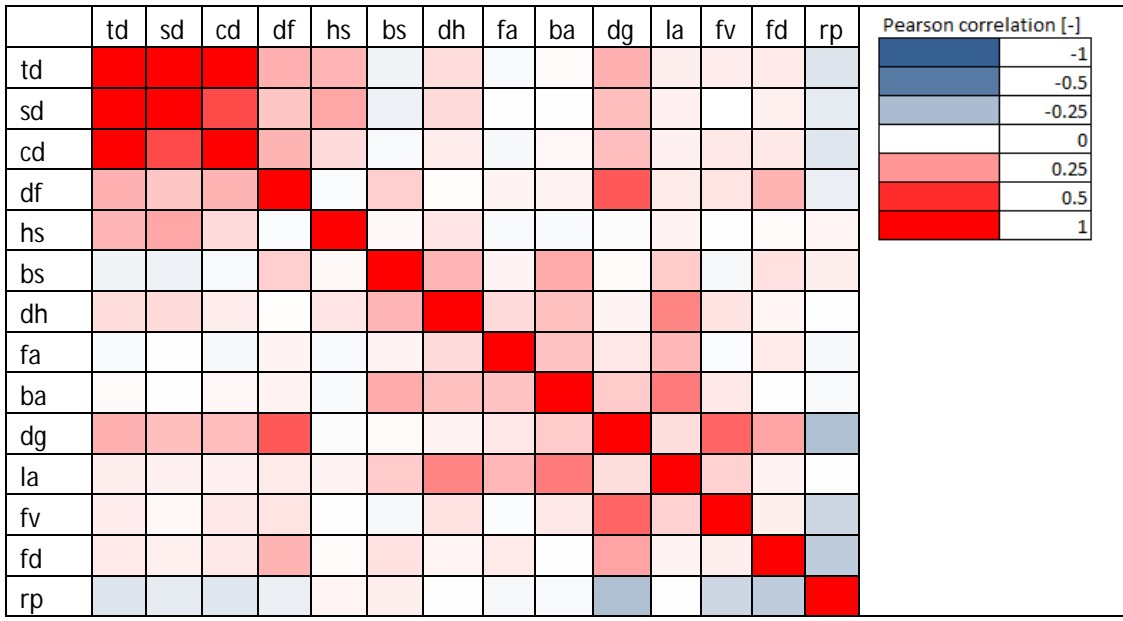

**Figure 3: Correlation coefficients between the different variables. See Table 1 for a description of the abbreviations).**

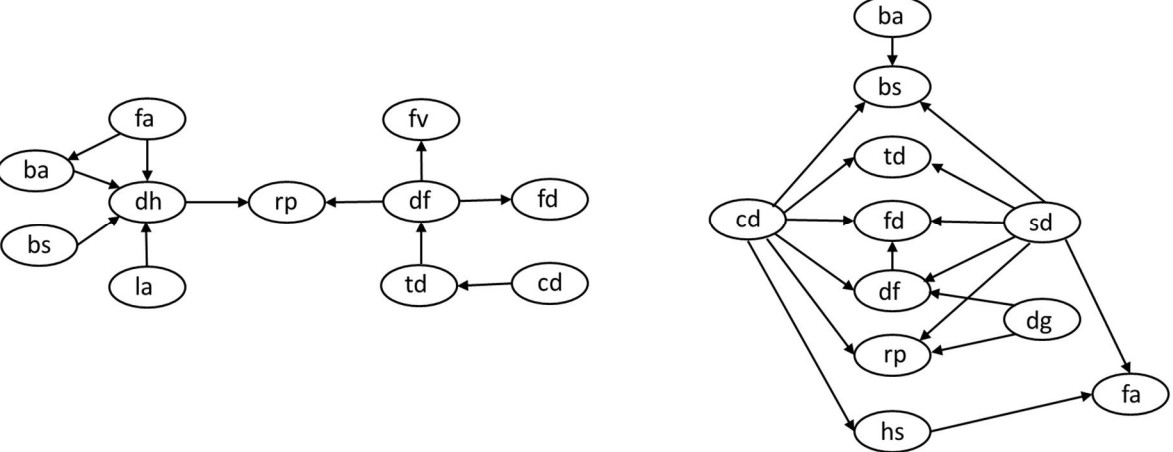

**Figure 4: Bayesian Network learned from data (left) and Bayesian Network constructed by experts (right). Note that not all variables are used in the network.**

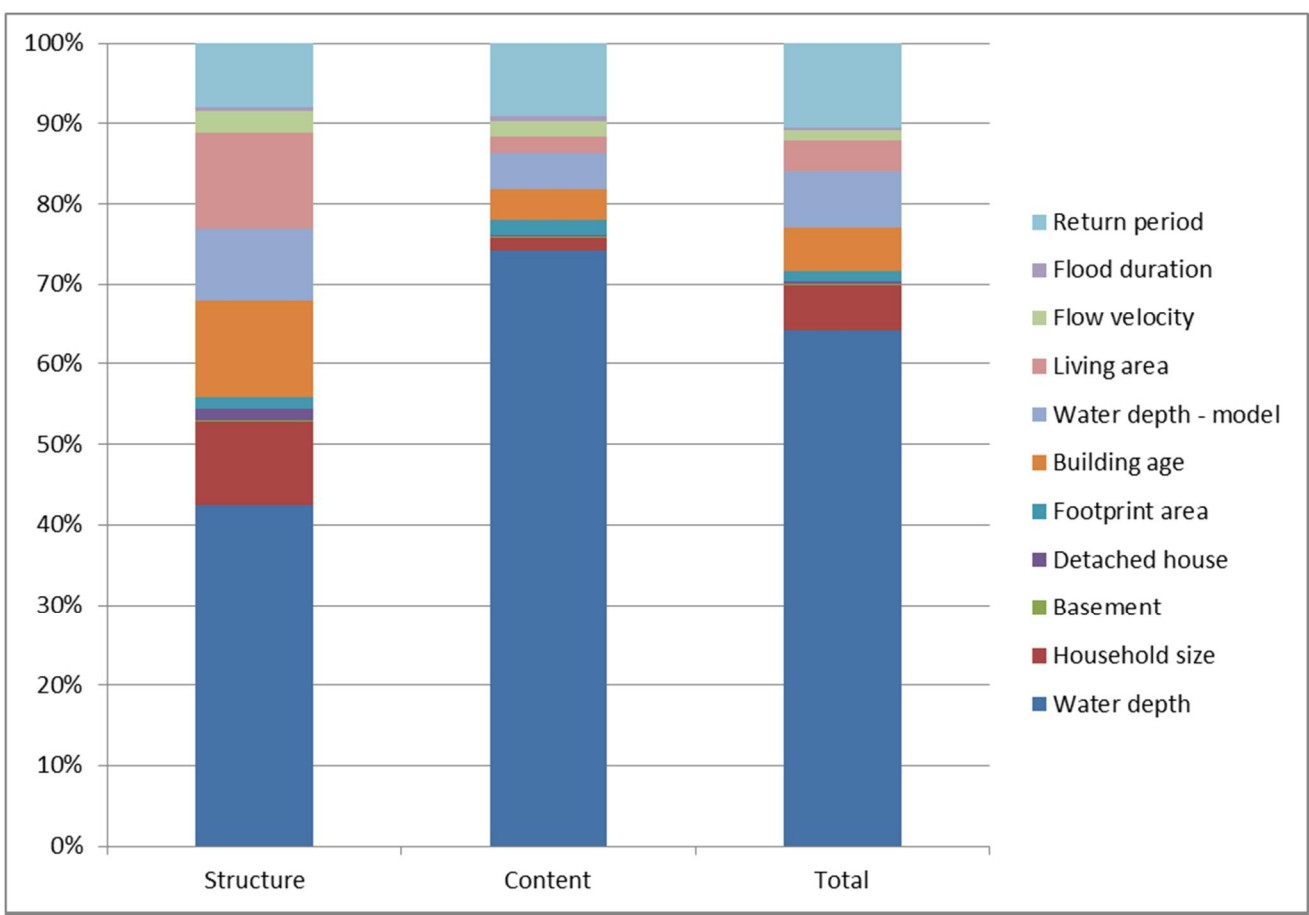

**Figure 5: Variable importance: The contribution of different variables in reducing the error in the bagging regression trees (the chart follows the order of the legend).**

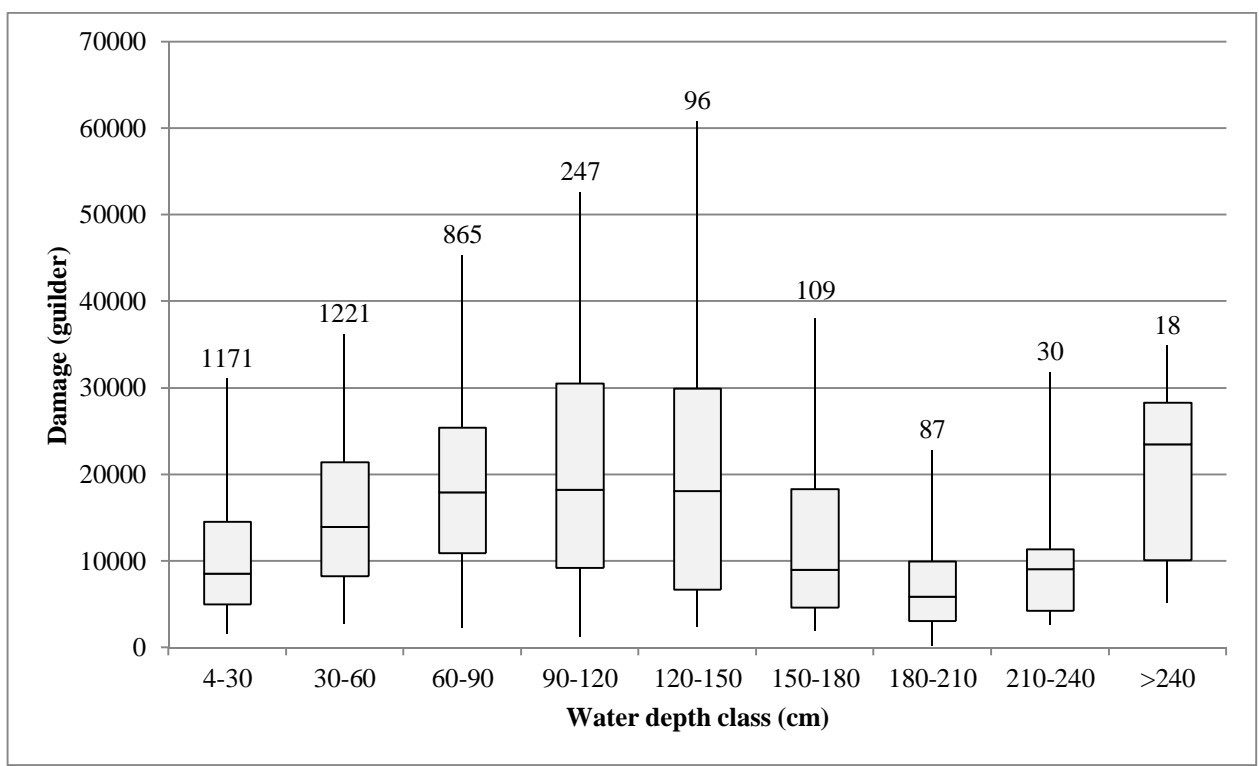

**Figure 6: Box-plots of the Meuse flood of 1993 per water depth class. The box shows the 25-75% interval and the lines show the 5-95% interval. The line in the middle of the box shows the median value. The labels on top of the plots show the number of observations per water depth class.**

