# Peer review of "Multi-variable flood damage modelling with limited data using supervised learning approaches"

_Natural Hazards and Earth System Sciences, 2017_

## Referee Comment (RC1) · Anonymous Referee #1 · 10 Feb 2017

The present manuscript about flood damage modeling with limited data pursues an interesting approach. The topic fits well to the scope of NHESS. However, there are several issues that need to be clarified or reconsidered:

**1   General Comments**

1. The authors may want to reconsider the title of the manuscript. "Data mining" is very prominent in the title, but I think this does not reflect the main focus of this paper. As a matter of fact, the aim of this manuscript is not primarily to do classical data mining on a huge data set (i.e. clustering, anomaly detection, classification), but rather to employ various unsupervised learning algorithms with the

aim of finding the best model to explain flood damage with a couple of independent variables (which of course is a part of data mining). Thus, the aim is to compare methodologies for a specific application example, rather than discovering patterns in a huge data set. To emphasize the focus of this work (multivariate flood damage modeling, limited data), I would suggest to rephrase the title to "Multi-variable flood damage modeling with limited data using supervised learning approaches."

2. Results and conclusions should be pointed out more clearly in the abstract. Please provide concise information regarding the improvements instead of pointing out a "significant improvement" and mentioning that some models "perform better".

3. With respect to the presentation quality, I advise to work on the language, on the structure of the manuscript and on the presentation of results. The "common thread" and the main take-home messages are not fully clear and concise throughout the manuscript. The discussion is relatively short, even though there are plenty of interesting aspects in this research that would be worth discussing, and that need to be discussed against the background of uncertain input data. Some formulations are too difficult to understand from a linguistic point of view. For instance p2, l23: "More commonly available (although still rare) are simple datasets that hold records with the flood damage that occurred for each building with sometimes a few other variables (such as location or water depth)."

4. The authors might want to think about the reference function. The root function is a simple, univariate function, which serves as a reference for sophisticated multivariate methods. Total damage and water depth are correlated with r = 0.18; I would assume that this value doesn't change significantly when calculating the correlation between total damage and the square root of the water depth. So the reference model is actually a rather bad model, possible improvements regarding

the GOF when using more advanced methods seem natural. It might be interesting to include a more sophisticated regression model as a reference (e.g. using LASSO, as this includes both variable selection and regularization).

5. Results with respect to the most important variables should be reported in greater detail. It is not clear to me which of the variables are actually beneficial for modeling total damage. The correlation coefficients in Table 1 do not provide any information on that, neither do the other tables or the results section. Variable selection is not discussed at all in this manuscript. For instance, total damage in the Bayesian networks is apparently (c.f. p25, l25-32; Figure 3) influenced by water depth (data-driven network) or water depth and structure damage (expert network), implying that there is no added value of using additional data. Table 4 somehow indicates that the increase in GOF is primarily dependent on the algorithms applied, rather than on additional data.

6. Given that the benefits of additional data are emphasized in this manuscript (p11, line 5; p11, line 26-27), it is worth discussing that care has to be taken when introducing additional data to a model. Even though pruning and bagging is mentioned, this topic is not really emphasized in this manuscript. Showing awareness of regularization and penalization of additional variables is of prime importance.

7. Uncertainty estimation is mentioned as one of the main merits of the methods used in this manuscript (p2, l16). However, uncertainties are neglected in the discussion section of this paper. Even though a number of sources for uncertainty are pointed out (e.g. data collected by different organizations; exact locations of buildings are not known; water depth is only based on estimates and has been questioned by experts; collection methods for variables "inhabitants", "basement" and "attached buildings" are unknown; uncertain join of data based on water depth rank), implications are not discussed.

8. Tables:

(a) Table 1: last column should read "Pearson correlation on total damage" (there are 3 different damage variables in the data set).

(b) Table 2: caption: "...algorithms". Column names for col 2 and 3 need to be more specific, "water depth" is also part of the "original variables". However, the authors may wish to reconsider if this table is really needed, all information presented is the text.

(c) Table 3: It is not clear to which dataset (water depth only / original data set / all variables) these values refer to. I guess it is the data set containing all variables, except for the root function? In addition, the authors may wish to consider adding a GOF-measure for the explained variation (i.e. $R^2$) to the table.

(d) Table 4: It is not really clear how the "best performing" models have been selected – seemingly on the correlation coefficient? Table 3 indicates that RMSE and correlation coefficient of bagging regression tree and random forest show almost identical GOF.

(e) May I propose to combine Tables 3 and 4 by reporting all values (i.e. RMSE, $r$, and maybe $R^2$ for all 6(5) methods, structured by input dataset). This would also incorporate the information from Table 2.

9. Figures:

(a) Figure 1: please rephrase caption, e.g. "Scatter plot showing the relation between water depth and damage in the original data set".

(b) Figure 2: It might be interesting to check if plotting the affected houses atop the water depth is easier to understand. It seems that some houses on the left map are not located in the inundated area at all, albeit they are labelled as "affected objects". A comparison is quite difficult, because the map sections are not identical (right map is shifted slightly towards northwest).

(c) Figure 3: The caption should be rephrased – td, sd and cd are no "predictors" (as mentioned at p5, l32).

(d) Figure 4: please rephrase the capture, e.g.: "Bayesian Network learned from data (left) and Bayesian Network constructed by experts (right). Note that not all variables are used in the network."

(e) Figure 5: The authors might reconsider plotting only the mean value for each class – boxplots for each bin would be more informative. Please include the number of observations to for each category.

10. Please adhere to the journal standards concerning references (see guidelines for authors). References should be formatted accordingly and consistently, and references should be sorted alphabetically.

11. References to relevant literature are sparse, while the number of references to gray literature is relatively high. Especially sections 1 and 2 would benefit from some additional references.

12. Please adhere to the journal standards concerning references (see guidelines for authors). References should be formatted accordingly and consistently, and references should be sorted alphabetically.

13. References to relevant literature are sparse, while the number of references to gray literature is relatively high. Especially sections 1 and 2 would benefit from some additional references.

**2 Specific Comments**

1. p1, line 8: "Flood damage assessment is usually done with damage curves only dependent on the water depth." – I would agree that most assessments include

water depth as the main determinant of direct damage, but against the background of recent research, I would disagree that it is still state of the art to build flood damage assessments solely on water depth (c.f. Dutta et al., 2003; Kreibich et al., 2005; Thieken et al., 2005; Apel et al., 2009; Elmer et al. 2010; Merz et al. 2013; van Ootegem et al. 2015; Gerl et al. 2016). I would advise to slightly rephrase this sentence, indicating that more sophisticated, multivariate approaches (including hydrological modeling) are on the rise.

2. p1, line 20–21: "Because flood risk management becomes increasingly risk-based, flood damage estimation is increasingly important in flood risk assessment." Please rephrase, this is unclear.

3. p1, line 23: "...flood risk assessments are..."

4. p1, line 27: "These models typically predict the fraction of damage..." – the authors may wish to clarify what the denominator of the fraction is by adding e.g. "...as percentage of total possible damage".

5. p2, line 11: "... based on a German dataset based on ..." – please rephrase.

6. p2, line 11ff: The authors might want to add some additional references to their literature overview about multi-variate flood damage models. In addition, it might be of interest for the reader to know about the types of covariates used in these studies.

7. p2, line 13: "Spekkers et al. (2014) did something similar ..." – please specify.

8. p2, line 14: "These multi-variable flood damage models have been shown to perform better..." – the authors may want to provide some quantitative indication regarding how much the performance of these multi-variate models exceeded the performance of simple flood damage models.

9. p2, line 27: "...that is used here, and previously described..." – please rephrase

10. p2, line 29: "...very different from the datasets used so far (fewer variables, different sources of variables and different country)." – please explain in more detail. What is meant by "different sources" and why is data from the Netherlands expected to be "very different" from data from Germany? Also, this seems to refer only to the data set used by Merz et al. (2013) and Vogel et al. (2014).

11. p3, line 11: "The dataset used in this study is based on..."

12. p3, line 12: "...in the Netherlands (WL Delft, 1994)."

13. p3, line 13: 180 km$^2$

14. p3, line 14: "32 % of the damage pertains to residential buildings and content, for this study only the damage to this category is used". – please rephrase.

15. p3, line 14: Please explain briefly why you decided not to consider damage to business and government buildings.

16. p3, line 17: I think the term "citizen household" is not very common. Maybe replace with "private households"?

17. p3, line 17ff: Please use a consistent, clear terminology. Distinguishing between "citizen households" (p3, l17), "companies" (p3, l18), "rental residential buildings" (p3, l21) "residential buildings" (p3, l25) and "rental houses" (p3, l26) and "privately owned residential buildings" (p3, l22) is confusing.

18. p3, line 20–23: "The building structure ... content for the same structure." – please rephrase these two sentences to make this more clear.

19. p3, line 23: What is meant by "building content"? Furnishings?

20. p3, line 25: "The dataset did not include the building structure damage to all rental houses" – It is not clear to me until now, if the data have simply been collected by two different companies (as p3, l17ff imply) or if these two companies have also collected different types of data? Based on the text I assumed that structural damage to rental houses has been collected by "Stitching Watersnood Bedrijven 1993"?

21. p3, line 27: "Several manual actions were undertaken..." – please explain/provide some insight into what type of actions this could have been.

22. p3, line 30–31: So, apparently the "manual actions" are not known at all. Please refer to possible impacts of these manual actions on the results in the discussion.

23. p4, line 6: as a matter of fact, this correlation between water depth and damage is almost negligible. Other studies have found more obvious relationships between water depth and damage (e.g. Merz et al., 2003; Pistrika et al., 2009; Prettenthaler et al., 2010). The assumption that water depth as the main determinant of direct damage does not seem to hold in this case. Please discuss possible reasons for this weak correlation in the discussion (is this only due to the questionable quality of the water depth data mentioned at p4, l4?).

24. p4, line 9: "However, this data is not described..."

25. p4, line 23: "... and 40 meters."

26. p5, line 1: The authors may wish to explain shortly how return levels are computed.

27. p5 line 5: I do not understand why Figure 2 would show that most of the area floods frequently. Isn't this just a map about water depth?

28. p5, line 17: "The method of joining cadastre objects with damage records within a postal code area based on water depth rank is error prone." – This is a quite straightforward approach, which is understandable given the lack of further information. However, this join is probably linked with relatively high uncertainty, depending on the spatial resolution of the DEM used and the (uncertain) expert estimation of water depth in the first place. It was mentioned that between 1 and 20 buildings share the same 6 digit postal code (p4, l2), so mismatches are likely to occur in postal codes with a larger number of buildings. The authors are probably right that houses within a postal code area are similar to some extent, but I am not sure if this is true for variables like "household size" or "floor area for living". Are water depths within a postal code area similar, too, or are the ranks clearly distinguishable? The problem in case of a large number of mismatches is, that this just seemingly increases precision of the analysis. It might be worth testing if results change when simply using a mean/median value for all buildings within one postal code.

29. p5, line 29: "Several data mining (sometimes called machine learning) ..." – please rephrase. Even though these are closely linked and often used as synonyms, data mining and machine learning are not exactly identical. Rather, machine learning is a sub-field of data mining, i.e. data mining is not only restricted to machine learning methods.

30. p5, line 31: "...based on all independent variables (thus excluding total, content and structure damage)." – please, rephrase. This might be confusing to some readers, as the BN (Figure 4; p9, l34) includes content damage and structure damage.

31. p6, line 6: "...because many damage functions in the literature have this shape". – please provide references, additional to Merz et al. (2012).

32. p6, line 8: may I suggest to use different variable names (variable names with

subscripts, e.g. $d_t$ for total damage) in the formula? df is a common abbreviation for degrees of freedom.

33. p6, line 8: "...to get the smallest possible error based on the total damage and water depth data. The optimization of the coefficients is done with the Python package SciPy" – please rephrase and clarify (e.g. "...are optimized using ordinary least squares estimation from the Python package SciPy").

34. p6, line 15: "However, it is more common to ..."

35. p6, line 19: "...with 11 variables for each damage record."

36. p6, line 21 "...reduces maximally..." replace with "...is minimized..."

37. p6, line 22 and p6, line 25: "...by calculating the MSE reduction for all..." and "...is the reduction in MSE of total damage ..." ("MSE error" is redundant).

38. p6, line 23: abbreviation MSE is already explained in p6, l20.

39. p6, l. 24ff: please try to integrate the formula and the explanation of variables more naturally into the flow text. The sentence "The regression tree... (Pedregosa et al. 2011)." might be added at the end of the page.

40. p7, line 10: "the Matlab Statistical Toolbox (Matlab website)" – replace with "Matlab's 'Statistics and Machine Learning Toolbox'"

41. p7, line 11: "Python libraries do not support pruning" – I think there are custom implementations of pruning in Python. The authors might want to look at sgenoud's fork of the scikit-learn package at github.

42. p7, line 11: "performance of pruning was similar" – can you provide some information about the method and results of the comparison?

43. p8, line 9: the authors might want to add references to these fields of application.

44. p8, line 27: please cite the URL as a normal reference, i.e. "All calculations were done using the Python library libpgm (Cabot 2012)."

45. p9, line 6: "...balance was found by trying several discretization resolutions in order to gain the best results." – please rephrase and add more concise information ("...trying several discretization results until the best solution was found based on xxxxx criterion")

46. p9, line 13: "This was done manually by varying the discretization of the important variables until the smallest error was found" – this is rather vague. What is "manually"? What do you mean by "important variables" and what is the "smallest error"?

47. p9, line 7–15: please make this paragraph more concise

48. p9, line 28–32: please rephrase, focus on methodology and advantages/disadvantages of a manually established network. Contributing experts other than the authors should be added in an "Acknowledgements" section rather than in the text.

49. p9, line 33 – p10, line 1: "The total damage is ... and the content damage." Please explain in more detail, this is not fully conclusive to me.

50. p10, line 5: please provide information about important independent variables within the results section.

51. p10, line 15 : "(with different better training data)" – please rephrase

52. p11, line 1: the authors may wish to put section 3.2 into the discussion section.

53. p11, line 7–8: "The relatively good performance ... is striking." – the authors may wish to replace "striking" with "worth noting". Actually, given the rather bad fit of the root function (as explained by the authors in the following paragraph) and the concern about overfitting with regression trees, I assume that both the authors and the reader would have expected this behavior, at least to some extent.

54. p11, line 10: I think boxplots would be a more informative representation for Figure 5. Also, the conclusion of a "downward slope" based only on the means for each class should be interpreted with care. It has to be noted that variance/number of outliers gets smaller for data points with water depth > 1.3 m.p11, line 12: This is an interesting peculiarity of this data set. While it seems to be plausible that preparedness effects might mitigate total damage (note the very weak correlation of -0.09 in this case), it is counter-intuitive that return period is negatively correlated with water depth. Basically, events associated with high return periods are rare events with high water depth, i.e. the higher the water depth, the greater the return period. Under the assumption that values for return periods are relatively homogeneous for the Meuse flood (which was one actual event with a certain return period), this would mean that areas with a high water depth get flooded more frequently *at relatively higher water depths*. Yet, I would assume that they get flooded more frequently, but at lower level. So, in the case of the Meuse flood, areas with high water depth showed lower return periods. Does this indicate possible inaccuracies of the flood return period maps?

55. p11, line 21: "...are different from each other in more ways than just the water depth" – please rephrase.

56. p11, line 22: While overfitting based on a single variable is a valid concern, concluding to use multiple variables to avoid overfitting might be erroneous if the use of extra variables is not penalized.

57. p11, line 25: rephrase as "Discussion and conclusion"

58. p11–p12: please work on the discussion section, a large portion of page 12 (l10-l26) is is mainly about potential advantages of BN that are not visible in the results of this study.

59. p12, line 15–17: "but what tree is uncertain" – please rephrase (two times).

60. p12, last paragraph: please rephrase your final conclusions, this is somewhat clumsy from a linguistic point of view; e.g. split the first sentence into two sentences at the third "and": "In this paper we utilized different data sources compared to previous studies to acquire this data and showed that also on this dataset the methods are beneficial, especially the tree-based methods" – simplify, rephrase; "One possible way forward is to..." replace with "Future work may include ..."; etc.

---

## Referee Comment (RC2) · Anonymous Referee #2 · 10 Feb 2017

The authors present a very interesting study on multi-variable flood damage modelling using data from the Netherlands. They use software-based statistical approaches to overcome the challenge of data scarceness in damage documentation, which is an important step towards an enhanced flood risk management. As such, the topic is of considerable interest to the readers of NHESS, and the manuscript should be considered for publication.

However, there are some shortcomings in the current version of the NHESSD paper which I will address below. These shortcomings should be considered by the authors before the manuscript may become acceptable for inclusion in NHESS.

First of all I suggest to change the title a bit since according to my opinion the term "data mining" is a bit misleading in comparison to the work undertaken in this paper.

[Figure]

What about just "multi-variable flood damage modelling with limited data"?

Second, I have the feeling that some of the existing (and relevant) literature on this topic is not included in the Introduction so far. It would be interesting to see more than the presented references to (mostly) Dutch researchers and the Potsdam group, e.g., by broadening the focus a bit towards works on flooding with sediment transport – here similar problems are described that somehow the deposition height is the only available parameter, but in turn this parameter is not fully representing the processes leading to loss. Examples include the works of Papathoma-Köhle or Fuchs, to just drop some names.

Third, in the discussion on vulnerability of buildings exposed to flood hazards there are some works not comparing direct losses, but the degree of loss, which is a relative measure taking into account the different building values. As such, and I am not completely familiar with Dutch building regulations, different loss heights are also a result of different values of the elements at risk. How did the authors consider this challenge during their analysis (which is also perfectly mirrored by Figure 1)?

Fourth, I kindly would like to suggest that the Results and Discussion (!) sections are more carefully written since so far, the first includes lots of discussion, and the current Conclusion and Discussion section is rather short. This should also include some paragraphs on the uncertainties behind the analysis, as mentioned in the Methods section.

Fifth I would like to recommend that the authors show a more detailed situation as the one presented in Figure 2 – the current scale is hardly readable. A possible solution is to show the overall extent as an inlet map and then in the main map just a zoom of the most interesting river section or so. For the legend: the water depths of 0.5, 1.0 and 2.0 m are not clearly distinguishable, and technically should be presented differently (e.g., by using the ">"). For some of the other Figs. presented I also would like to recommend to clearly state the abbreviations (e.g., td, sd, cd,...) in the Figure caption.

Finally, I would recommend to extent the discussion on Fig. 5 – as already indicated there may be variables other than the water height responsible for the loss height available...

I strongly encourage the authors to perform towards the suggestions since the work presented is of particular interest and importance to the flood hazard community. I am looking forward to review a revised version.

---

## Author Comment (AC1) · 21 Apr 2017

**Response to review Referee 1:**

Thank you very much for your helpful and detailed comments and suggestions. The number of helpful suggestions, and also detailed comments on the text, is great and much appreciated. They contribute a lot to improving our manuscript. Below, we respond to the individual comments.

1.  *The authors may want to reconsider the title of the manuscript. "Data mining" is very prominent in the title, but I think this does not reflect the main focus of this paper. As a matter of fact, the aim of this manuscript is not primarily to do classical data mining on a huge data set (i.e. clustering, anomaly detection, classification), but rather to employ various unsupervised learning algorithms with the aim of finding the best model to explain flood damage with a couple of independent variables (which of course is a part of data mining). Thus, the aim is to compare methodologies for a specific application example, rather than discovering patterns in a huge data set. To emphasize the focus of this work (multivariate flood damage modeling, limited data), I would suggest to rephrase the title to "Multi-variable flood damage modeling with limited data using supervised learning approaches."*

    We agree that the word "Data-Mining" is broad and also covers many things not done in this paper. Our motivation for using it was to follow the terminology used in Merz et al. (2013), an early publication about the application of supervised learning algorithms in flood damage estimation. However, considering the comments about this by both referees and that we actually agree that "Data-Mining" is a very broad term; we propose to change the title as proposed in this review. The new title will be: "Multi-variable flood damage modeling with limited data using supervised learning approaches".

2.  *Results and conclusions should be pointed out more clearly in the abstract. Please provide concise information regarding the improvements instead of pointing out a "significant improvement" and mentioning that some models "perform better".*

    We will mention the improvements in the goodness of fit (GOF) values in both the abstract and the conclusions. Also we will mention the exact differences in performance among the different models.

3.  *With respect to the presentation quality, I advise to work on the language, on the structure of the manuscript and on the presentation of results. The "common thread" and the main takehome messages are not fully clear and concise throughout the manuscript. The discussion is relatively short, even though there are plenty of interesting aspects in this research that would be worth discussing, and that need to be discussed against the background of uncertain input data. Some formulations are too difficult to understand from a linguistic point of view. For instance p2, l23: "More commonly available (although still rare) are simple datasets that hold records with the flood damage that occurred for each building with sometimes a few other variables (such as location or water depth)."*

    We will go again through the manuscript and look critical at the language. Also we will add more text to the discussion (see more about changes to the discussion section, in our response to referee 2).

4. *The authors might want to think about the reference function. The root function is a simple, univariate function, which serves as a reference for sophisticated mul-tivariate methods. Total damage and water depth are correlated with r = 0.18; I would assume that this value doesn't change significantly when calculating the correlation between total damage and the square root of the water depth. So the reference model is actually a rather bad model, possible improvements regarding the GOF when using more advanced methods seem natural. It might be interesting to include a more sophisticated regression model as a reference (e.g. using LASSO, as this includes both variable selection and regularization).*

The purpose of the reference model is to compare the supervised learning algorithms with what is currently typically applied in flood risk management studies. Very few studies (outside academic research) already apply multi-variable functions and therefore we chose a uni-variate function as a reference. Secondly, many expert damage functions look like a square root function. We therefore believe this is a relevant reference model as it represents what is commonly applied. However, we agree that it is a poor alternative to the techniques later applied in this research. We therefore will add the LASSO technique as a second reference.

5. *Results with respect to themost important variables should be reported in greater detail. It is not clear to me which of the variables are actually beneficial for modeling total damage. The correlation coefficients in Table 1 do not provide any information on that, neither do the other tables or the results section. Variable selection is not discussed at all in this manuscript. For instance, total damage in the Bayesian networks is apparently (c.f. p25, l25-32; Figure 3) influenced by water depth (data-driven network) or water depth and structure damage (expert network), implying that there is no added value of using additional data. Table4 somehow indicates that the increase in GOF is primarily dependent on the algorithms applied, rather than on additional data.*

Variable selection was not one of the initial purposes of this manuscript. The referee is therefore right that no information about it can be found in the manuscript. However, we agree that variable selection provides some important extra information regarding the benefits of extra data. It is therefore very relevant to this manuscript and therefore will add it to the revised manuscript. We will use out-of-bag techniques to say something about variable importance.

6. *Given that the benefits of additional data are emphasized in this manuscript (p11, line 5; p11, line 26-27), it is worth discussing that care has to be taken when introducing additional data to a model. Even though pruning and bagging is mentioned, this topic is not really emphasized in this manuscript. Showing awareness of regularization and penalization of additional variables is of prime importance.*

We fully agree with the referee that potential overfitting is an essential topic (the mentioned methods, regularization, penalization, bagging and pruning, are all methods to avoid overfitting). In the study a lot of care went into avoiding overfitting of the tree based

methods. Furthermore, our testing set was not used for training the models, so problems with overfitting would come back in the GOF indicators. If, for example, the different methods to avoid overfitting would not be used on the regression trees, the GOF would be much worse on the testing data and nearly perfect on the training data. Below, we describe the considerations that we will clarify in the revised paper on the different statistical approaches:

**Tree based models**

For the tree based methods we avoided overfitting with a minimum data required per tree leaf combined with a maximum number of splits per tree. For the simple regression tree method we compared this technique with pruning and found similar results. The random forest and bagging tree methods are in itself already less sensitive to overfitting, however the same techniques were applied to avoid overfitting. Regularization is not commonly used to avoid overfitting in tree based methods.

**Bayesian Network**

Overfitting was not addressed in the manuscript. The assumption was that Bayesian Networks are not very prone to overfitting and also the applied library has no settings to avoid overfitting. This assumption seems correct. If instead of the separate test set the Bayesian Network is tested on the learning data, the GOF indicators show no improvement. If overfitting would be a problem the GOF indicators would be much better when the training data is used for testing. This line of argumentation will be added to the Bayesian Network section.

7. *Uncertainty estimation is mentioned as one of the main merits of the methods used in this manuscript (p2, l16). However, uncertainties are neglected in the discussion section of this paper. Even though a number of sources for uncertainty are pointed out (e.g. data collected by different organizations; exact locations of buildings are not known; water depth is only based on estimates and has been questioned by experts; collection methods for variables "inhabitants", "basement" and "attached buildings" are unknown; uncertain join of data based on water depth rank), implications are not discussed.*

Uncertainty estimation was not a purpose of this manuscript, however, uncertainty is an important issue in flood damage estimation and some of the methods applied could help quantify the uncertainty. That is why the uncertainty estimation qualities of the different techniques were presented in the discussion section. The uncertainty in the input data is discussed throughout the manuscript. In the discussion we will add an overview of these uncertainties and discuss the implications.

8. *(a) Table 1: last column should read "Pearson correlation on total damage"(there are 3 different damage variables in the data set).*
*(b) Table 2: caption: "...algorithms". Column names for col 2 and 3 need to be more specific, "water depth" is also part of the "original variables". However, the authors may wish to reconsider if this table is really needed, all information presented is the text.*
*(c) Table 3: It is not clear to which dataset (water depth only / original data set / all variables) these values refer to. I guess it is the data set containing all variables, except for the root*

*function? In addition, the authors may wish to consider adding a GOF-measure for the explained variation (i.e. R2) to the table.*
*(d) Table 4: It is not really clear how the "best performing" models have been selected – seemingly on the correlation coefficient? Table 3 indicates that RMSE and correlation coefficient of bagging regression tree and random forest show almost identical GOF.*
*(e) May I propose to combine Tables 3 and 4 by reporting all values (i.e. RMSE, r, and maybe R2 for all 6(5) methods, structured by input dataset). This would also incorporate the information from Table 2.*

A) Agree, will be done. B) Agree, we will omit this table. C) We will clarify this table and add the R2 indicator. D) The best performing method has been selected based on a combination of the GOF indicators, in case of similar results only one was picked. This will be clarified in the revised paper and in the cases of similar results this will be mentioned. E) We will add the R2 GOF to both tables. Merging the tables is possible but we feel that this is not very practical, as it would result in one complex table rather than two simple ones. Now the tables neatly address a different question (best performing algorithm; and improvements when more data is added).

9. *(a) Figure 1: please rephrase caption, e.g. "Scatter plot showing the relation between water depth and damage in the original data set".*
*(b) Figure 2: It might be interesting to check if plotting the affected houses atop the water depth is easier to understand. It seems that some houses on the left map are not located in the inundated area at all, albeit they are labelled as "affected objects". A comparison is quite difficult, because the map sections are not identical (right map is shifted slightly towards north-west).*
*(c) Figure 3: The caption should be rephrased – td, sd and cd are no "predictors" (as mentioned at p5, l32).*
*(d) Figure 4: please rephrase the capture, e.g.: "Bayesian Network learned from data (left) and Bayesian Network constructed by experts (right). Note that not all variables are used in the network."*
*(e) Figure 5: The authors might reconsider plotting only themean value for each class – boxplots for each bin would be more informative. Please include the number of observations to for each category.*

A) We will rephrase it to the suggested phrase. B) It is correct that the left map is a map of all objects in the 1993 situation rather than the affected objects. We can plot them on top of each other only showing the actually affected objects. C) We will change the word "predictors" to "variables". D) We will rephrase this in the suggested way. E) We will use boxplots and add the number of observations.

10. *Please adhere to the journal standards concerning references (see guidelines for authors). References should be formatted accordingly and consistently, and references should be sorted alphabetically.*

We will improve the reference list according to the journal standards.

11. *References to relevant literature are sparse, while the number of references to gray literature is relatively high. Especially sections 1 and 2 would benefit from some additional references.*

   We will add some key references to peer-reviewed papers in the introduction section, and to the section describing the different algorithms. For the description of the dataset there is unfortunately mostly gray literature available, except for the paper by Wind et al. (1999) in GRL which we already quote.

12. *Please adhere to the journal standards concerning references (see guidelines for authors). References should be formatted accordingly and consistently, and references should be sorted alphabetically.*

   See 10

13. *References to relevant literature are sparse, while the number of references to gray literature is relatively high. Especially sections 1 and 2 would benefit from some additional references.*

   See 11

Specific comments:

1. *p1, line 8: "Flood damage assessment is usually done with damage curves only dependent on the water depth." – I would agree that most assessments include water depth as the main determinant of direct damage, but against the background of recent research, I would disagree that it is still state of the art to build flood damage assessments solely on water depth (c.f. Dutta et al., 2003; Kreibich et al., 2005; Thieken et al., 2005; Apel et al., 2009; Elmer et al. 2010; Merz et al. 2013; van Ootegem et al. 2015; Gerl et al. 2016). I would advise to slightly rephrase this sentence, indicating that more sophisticated, multivariate approaches (including hydrological modeling) are on the rise.*

   A distinction should be made here between the scientific literature and actual flood risk management studies. We will add a sentence that multi-variable approaches have been carried out in recent academic research, especially in Germany.

2. *p1, line 20–21: "Because flood risk management becomes increasingly risk-based, flood damage estimation is increasingly important in flood risk assess-ment." Please rephrase, this is unclear.*

   We will rephrase it into: "Decision making in flood risk management is increasingly based on studies that quantify the flood risk rather than only the flood hazard. Flood damage estimation is therefore increasingly important."

3. *p1, line 23: "...flood risk assessments are..."*

   We will change this.

4. *p1, line 27: "These models typically predict the fraction of damage..." – the authors may wish to clarify what the denominator of the fraction is by adding e.g. "...as percentage of total possible damage".*

We will add "as percentage of potential damage"

5. *p2, line 11: "... based on a German dataset based on ..." – please rephrase.*

We will change it into "with a German dataset based on .."

6. *p2, line 11ff: The authors might want to add some additional references to their literature overview about multi-variate flood damage models. In addition, it might be of interest for the reader to know about the types of covariates used in these studies.*

We will reference to the suggested literature of point 1 and discuss the methods they applied.

7. *p2, line 13: "Spekkers et al. (2014) did something similar ..." – please specify.*

We will change this in: "Spekkers et al. (2014) applied regression trees to estimate pluvial flood damage".

8. *p2, line 14: "These multi-variable flood damage models have been shown to perform better..." – the authors may want to provide some quantitative indication regarding how much the performance of these multi-variate models exceeded the performance of simple flood damage models.*

We will add some GOF values from Schröter et al. (2014).

9. *p2, line 27: "...that is used here, and previously described..." – please rephrase*

We will change this into: "...which is used here. Previously this dataset has been described in Wind et al. (1999) and in more detail in WL Delft (1994).

10. *p2, line 29: "...very different from the datasets used so far (fewer variables, different sources of variables and different country)." – please explain in more detail. What is meant by "different sources" and why is data from the Netherlands expected to be "very different" from data from Germany? Also, this seems to refer only to the data set used by Merz et al. (2013) and Vogel et al. (2014).*

With different source we mean that the data was collected by insurance experts directly after the floods for compensation purposes and covers all affected buildings. This is different from the German data which was collected a year after the flood for research purposes based on a sample of the affected buildings. The data is also different in that in the original study only a few variables were collected, most of the other variables are added

later. In contrast for the German dataset all variables (except return period) were based on telephone interview answers. A few studies also applied datasets different from the GFZ data. These studies did however use different analysis methods (e.g. Dutta et al. 2003) or focused on pluvial flood damage (Van Oostegem et al., 2015 and Spekkers et al., 2014). We will add this explanation to the revised manuscript.

11. *p3, line 11: "The dataset used in this study is based on..."*

    We will change this as suggested.

12. *p3, line 12: "...in the Netherlands (WL Delft, 1994)."*

    We will change this as suggested.

13. *p3, line 13: 180 km$^2$*

    We will change this as suggested.

14. *p3, line 14: "32% of the damage pertains to residential buildings and content, for this study only the damage to this category is used". – please rephrase.*

    We will change this into: "…32% of the damage pertains to residential buildings and content. In this study only residential damage is considered. "

15. *p3, line 14: Please explain briefly why you decided not to consider damage to business and government buildings.*

    We will add the sentence: "Other damage categories are not considered because they are more heterogeneous and less data about them is available."

16. *p3, line 17: I think the term "citizen household" is not very common. Maybe replace with "private households"?*

    See 17.

17. *p3, line 17ff: Please use a consistent, clear terminology. Distinguishing between "citizen households" (p3, l17), "companies" (p3, l18), "rental residential buildings" (p3, l21) "residential buildings" (p3, l25) and "rental houses" (p3, l26) and "privately owned residential buildings" (p3, l22) is confusing.*

    We will change: "rental houses" to "rental residential buildings" and "citizen households" to "privately owned residential buildings".

18. *p3, line 20–23: "The building structure ... content for the same structure." –please rephrase these two sentences to make this more clear.*

We will rephrase this into: "In this set up of the damage collection, the building structure of rental residential buildings was collected by "Stichting Watersnood bedrijven", the organization that collected company damages. This is different from the organization that collected the rest of the residential damages. From the company damages less information was shared to WL Delft (1994), the source of the dataset for this study."

19. *p3, line 23: What is meant by "building content"? Furnishings?*

Building content is a commonly applied term in flood damage literature (both German and US studies use it consistently). However, the reviewer is right that there are UK studies that apply the word furnishings instead. At the first mentioning of the word building content we will add the word furnishings between brackets.

20. *p3, line 25: "The dataset did not include the building structure damage to all rental houses" – It is not clear to me until now, if the data have simply been collected by two different companies (as p3, l17ff imply) or if these two companies have also collected different types of data? Based on the text I assumed that structural damage to rental houses has been collected by "Stiching Watersnood Bedrijven 1993"?*

We will clarify this already a bit earlier (see point 18). Our source for the data, the WL Delft report (1994) combined two different sources for the building data, and in one source (rental houses) the structure damage was available only in some unknown aggregate form. Probably because the rental residential building damage was collected per owner and one owner could own multiple buildings. This reason is however speculation and was therefore not mentioned in the manuscript. The bottom line is that the sum of all building values is known (we verified this with Wind et al., (1999)), but that the distribution of this value over individual objects is uncertain for a part of the structure damage. The share of rental buildings is however expected to be low in this rural area and therefore we expect this to not substantially affect our results. We will mention these issues more explicitly in the revised paper.

21. *p3, line 27: "Several manual actions were undertaken…" – please explain/provide some insight into what type of actions this could have been.*

We speculate that the organization collecting the data had information on the total structural damage to rental houses, and divided this over the rental objects, based on the number of inhabitants. We will add this in the revised paper.

22. *p3, line 30–31: So, apparently the "manual actions" are not known at all. Please refer to possible impacts of these manual actions on the results in the discussion.*

We will add a paragraph about this in the discussion.

23. *p4, line 6: as a matter of fact, this correlation between water depth and damage is almost negligible. Other studies have found more obvious relationships between water depth and damage (e.g. Merz et al., 2003; Pistrika et al., 2009; Prettenthaler et al., 2010). The assumption that water depth as the main determinant of direct damage does not seem to hold in this case. Please discuss possible reasons for this weak correlation in the discussion (is this only due to the questionable quality of the water depth data mentioned at p4, l4?).*

Given our large dataset size (about 4000+ records) the correlation with water depth is not high, but not negligible either. The correlation coefficient is about half of what other studies found, however in looking at the variable importance we see that water depth is still by far the most important variable. The point of this study is that even though the correlation coefficient is weak and the dataset has some issues (as in many other cases around the world), we can still get significantly better damage estimates with this "limited data". We will emphasize this a bit more in the abstract and the conclusion of the paper. However, our estimates have not improved as much as one would have hoped, and this point will also be added to the discussion.

24. *p4, line 9: "However, this data is not described..."*

We will change this as suggested.

25. *p4, line 23: "... and 40 meters."*

We will change this as suggested.

26. *p5, line 1: The authors may wish to explain shortly how return levels are computed.*

P5 line 1-6 already contains this explanation. We will clarify that this return period here differs from the return period variable in the GFZ dataset, in that we use the return period for any flood at the specific object location and not the return period of the flood that actually occurred. Our hope is that this return period is a good proxy for flood experience of the population, while in the GFZ dataset it says something about the magnitude of the flood. This context will be added before the explanation of how the return periods are determined to clarify the explanation to the reader.

27. *p5 line 5: I do not understand why Figure 2 would show that most of the area floods frequently. Isn't this just a map about water depth?*

Correct, we removed the information on flood frequency from the draft paper, but the text remained. However given the previous comments on the return period variable (comment 26), we will add the map again to the revised paper so that the reader has a better understanding of the meaning of this variable.

28. *p5, line 17: "The method of joining cadastre objects with damage records within a postal code area based on water depth rank is error prone." – This is a quite straightforward*

*approach, which is understandable given the lack of further information. However, this join is probably linked with relatively high uncertainty, depending on the spatial resolution of the DEM used and the (uncertain) expert estimation of water depth in the first place. It was mentioned that between 1 and 20 buildings share the same 6 digit postal code (p4, l2), so mismatches are likely to occur in postal codes with a larger number of buildings. The authors are probably right that houses within a postal code area are similar to some extent, but I am not sure if this is true for variables like "household size" or "floor area for living". Are water depths within a postal code area similar, too, or are the ranks clearly distinguishable? The problem in case of a large number of mismatches is, that this just seemingly increases precision of the analysis. It might be worth testing if results change when simply using a mean/median value for all buildings within one postal code.*

This suggestion is appreciated, and we will perform this test.

29. *p5, line 29: "Several data mining (sometimes called machine learning) …" – please rephrase. Even though these are closely linked and often used as synonyms, data mining and machine learning are not exactly identical. Rather, machine learning is a sub-field of data mining, i.e. data mining is not only restricted to machine learning methods.*

As mentioned in point 1 of the detailed comments, we will remove the word data-mining from the manuscript and apply the term "supervised learning" instead, as helpfully suggested by the reviewer.

30. *p5, line 31: "…based on all independent variables (thus excluding total, content and structure damage)." – please, rephrase. This might be confusing to some readers, as the BN (Figure 4; p9, l34) includes content damage and structure damage.*

We will omit the part "based on all independent variables (thus excluding total, content and structure damage). ". This addition is indeed so obvious that it can be confusing.

31. *p6, line 6: "…because many damage functions in the literature have this shape".– please provide references, additional to Merz et al. (2012).*

In Wagenaar et al. (2016) there is a figure with damage functions from different studies. Most of the damage functions have approximately the root function shape. For example, HAZUS (Scawthorn et al. 2006), MCM (Penning-Roswell, 2005), Tebodin (Sluijs et al., 2000) and Flemo (Thieken et al., 2008). We will add this to the revised paper.

32. *p6, line 8: may I suggest to use different variable names (variable names withsubscripts, e.g. dt for total damage) in the formula? df is a common abbreviation for degrees of freedom.*

We thank the reviewer for the suggestion to use more consistent abbreviations. df stands for "depth relative to floor", we will change this into "wdf", adding the "w" of water. The use of sub-scripts then would not be necessary.

33. *p6, line 8: "...to get the smallest possible error based on the total damage and water depth data. The optimization of the coefficients is done with the Python package SciPy" – please rephrase and clarify (e.g. "...are optimized using ordinary least squares estimation from the Python package SciPy").*

   We will change this into: "The values of the coefficients are optimized for the best fit with the ordinary least squares method. This is done with the Python package SciPy.

34. *p6, line 15: "However, it is more common to ..."*

   We will change this as suggested.

35. *p6, line 19: "...with 11 variables for each damage record."*

   We will change this as suggested.

36. *p6, line 21 "...reduces maximally..." replace with "...is minimized..."*

   We will change this as suggested.

37. *p6, line 22 and p6, line 25: "...by calculating the MSE reduction for all..." and "...is the reduction in MSE of total damage ..." ("MSE error" is redundant).*

   We will remove the second "MSE error"

38. *p6, line 23: abbreviation MSE is already explained in p6, l20.*

   We will just use the abbreviation here.

39. *p6, l. 24ff: please try to integrate the formula and the explanation of variables more naturally into the flow text. The sentence "The regression tree... (Pedregosa et al. 2011)." might be added at the end of the page.*

   We will rewrite this part as suggested.

40. *p7, line 10: "the Matlab Statistical Toolbox (Matlab website)" – replace with "Matlab's 'Statistics and Machine Learning Toolbox"'*

   We will change this as suggested.

41. *p7, line 11: "Python libraries do not support pruning" – I think there are custom implementations of pruning in Python. The authors might want to look at sgenoud's fork of the scikit-learn package at github.*

The main version of Scikit learn (well-known Machine Learning library in Python) doesn't support pruning. An internet search didn't yield any alternative in major libraries that do support this. With some effort we might be able to find and use a GitHub implementation of pruning in Python by someone. However, we did successfully run the pruning algorithm in Matlab and found no better results than in Python without pruning (and using alternative methods to avoid overfitting). Furthermore, pruning is mostly relevant for traditional regression trees and not for Random Forests or Bagging trees. Traditional regression trees are currently far from the best performing algorithm and including pruning is therefore not expected to influence the conclusions of this paper in any way. We therefore would not apply a Python implementation of pruning for this paper.

42. *p7, line 11: "performance of pruning was similar" – can you provide some information about the method and results of the comparison?*

We compared them based on the RMSE indicator. We will run the Matlab script again to get the exact values, and report these in the revised paper.

43. *p8, line 9: the authors might want to add references to these fields of application.*

We will look up references for applications in the different fields.

44. *p8, line 27: please cite the URL as a normal reference, i.e. "All calculations were done using the Python library libpgm (Cabot 2012)."*

We will change this as suggested.

45. *p9, line 6: "...balance was found by trying several discretization resolutions in order to gain the best results." – please rephrase and add more concise information ("...trying several discretization results until the best solution was found based on xxxxx criterion")*

We will rewrite this. The criterion was the RMSE. We changed the number of bins the different variables are divided in, and calculated after each change the RMSE. We then applied the number of bins with the smallest RMSE. This action was done by hand and not by an algorithm (hence manual).

46. *p9, line 13: "This was done manually by varying the discretization of the important variables until the smallest error was found" – this is rather vague. What is "manually"? What do you mean by "important variables" and what is the "smallest error"?*

See 45.

47. *p9, line 7–15: please make this paragraph more concise*

We will rewrite that paragraph in a more clear and structured manner. The content of the paragraph is however highly relevant and the rewrite will focus on style only.

48. *p9, line 28–32: please rephrase, focus on methodology and advantages/disadvantages of a manually established network. Contributing experts other than the authors should be added in an "Acknowledgements" section rather than in the text.*

We will add a more detailed discussion on the advantages/disadvantages of expert networks versus learned networks. The main advantages of an expert network are that the overfitting problem is less relevant and that experts take into account the variables/connections that are practically important. Advantage of a learned network are that new and previously unknown relationships between variables can be discovered. Also, we will add the experts contributing to the expert network (and not being authors of the paper) to the acknowledgements section, as suggested.

49. *p9, line 33 – p10, line 1: "The total damage is … and the content damage." Please explain in more detail, this is not fully conclusive to me.*

We will rephrase this into: "The relationship between the total damage, structural damage and content damage is known and not probabilistic: total damage = structure damage + content damage. Also, in our case the structure damage, content damage and total damage are always all dependent. Therefore, using a Bayesian Network to model this exact definitional relationship could only introduce extra errors and not add anything extra explanation."

50. *p10, line 5: please provide information about important independent variables within the results section.*

We will add this analysis. See point 5 of the main points.

51. *p10, line 15 : "(with different better training data)" – please rephrase*

We will remove the section between brackets and replace this with a new sentence. "Schröter et al. (2014) used another dataset, with more variables per damage record and applied more reliable collection methods".

52. *p11, line 1: the authors may wish to put section 3.2 into the discussion section.*

This is a good suggestion; we will split paragraph 3.2, the first paragraph and the table will remain in section 3. The second and third paragraph will be moved to the discussion.

53. *p11, line 7–8: "The relatively good performance … is striking." – the authors may wish to replace "striking" with "worth noting". Actually, given the rather bad fit of the root function (as explained by the authors in the following paragraph) and the concern about overfitting with regression trees, I assume that both the authors and the reader would have expected this behavior, at least to some extent.*

We will use the word "worth noting" instead of "striking". The bad fit of the root function is most unexpected here, given that most damage functions look like root functions and given that root functions are a logical relationship between water depth and damage. Our initial thought was that the root function performed bad because it only had the water depth as input. This expectation turned out wrong after seeing the good performance of the regression tree with the same information. At the description of the root function, additional argumentation for expecting a root function will be provided.

54. *p11, line 10: I think boxplots would be a more informative representation for Figure 5. Also, the conclusion of a "downward slope" based only on the means for each class should be interpreted with care. It has to be noted that variance/number of outliers gets smaller for data points with water depth > 1.3m.p11, line 12: This is an interesting peculiarity of this data set. While it seems to be plausible that preparedness effects might mitigate total damage (note the very weak correlation of -0.09 in this case), it is counter-intuitive that return period is negatively correlated with water depth. Basically, events associated with high return periods are rare events with high water depth, i.e. the higher the water depth, the greater the return period. Under the assumption that values for return periods are relatively homogeneous for the Meuse flood (which was one actual event with a certain return period), this would mean that areas with a high water depth get flooded more frequently at relatively higher water depths. Yet, I would assume that they get flooded more frequently, but at lower level. So, in the case of the Meuse flood, areas with high water depth showed lower return periods. Does this indicate possible inaccuracies of the flood return period maps?*

The reviewer misunderstood the meaning of our variable "return period". In this study we used the flood return period at a location for any flood not the return period of the flood that actually occurred. Therefore, there is a lot of variation in the return period variable within our dataset. In point 26 and 27 we propose ways to avoid this confusion. The reviewer is correct that the number of observations in figure 5 is smaller at the higher water depths (however the number of observations remain large). As suggested we will add box plots to solve this problem in figure 5. As for the possible inaccuracies in the flood return period map, these are expected to be insignificant. The absolute values might be inaccurate but in relative terms the return periods are expected to be good (low areas near the river have a frequent return period, high areas far from the river have an infrequent return period). For our study only the relative return periods are important. We will add a return period map so the reader can see that the return periods make sense.

55. *p11, line 21: "...are different from each other in more ways than just the water depth" – please rephrase.*

We will rephrase this into: "it shows that there are relevant differences between floods that cannot be expressed with the water depth variable alone".

56. *p11, line 22: While overfitting based on a single variable is a valid concern, con-cluding to use multiple variables to avoid overfitting might be erroneous if the use of extra variables is not penalized.*

    In fact it's not the overfitting on the event that is an argument for multi-variable damage functions; it's the physically unrealistic downward sloping damage function itself. This downward sloping could only occur if some other factor plays an important role. We will rephrase this part of the paper to make this clear. For the penalization see point 6 of the main points.

57. *p11, line 25: rephrase as "Discussion and conclusion"*

    We will rewrite this part as suggested.

58. *p11–p12: please work on the discussion section, a large portion of page 12 (l10-l26) is is mainly about potential advantages of BN that are not visible in the results of this study.*

    We will shorten the section about potential advantages of Bayesian networks not shown in this manuscript. Many points will be added to the discussion section, see points: 7,21,23 and our response to the comments from the other referee.

59. *p11–p12: please work on the discussion section, a large portion of page 12 (l10-l26) is is mainly about potential advantages of BN that are not visible in the results of this study.*

    We will rephrase into: "..but what tree is the correct tree is uncertain".

60. *p12, last paragraph: please rephrase your final conclusions, this is somewhat clumsy from a linguistic point of view; e.g. split the first sentence into two sentences at the third "and": "In this paper we utilized different data sources compared to previous studies to acquire this data and showed that also on this dataset the methods are beneficial, especially the tree-based methods" – simplify, rephrase; "One possible way forward is to..." replace with "Future work may include ..."; etc.*

    We revise the first sentence in this paragraph using the suggestions of the reviewer.

**Review referee 2:**

Thank you very much for your thoughtful and interesting comments. They contribute a lot to further improving our manuscript better.

1. *First of all I suggest to change the title a bit since according to my opinion the term "data mining" is a bit misleading in comparison to the work undertaken in this paper. What about just "multi-variable flood damage modelling with limited data"?*

We agree with the referees comment about the title. This is addressed in detail in our comment to point 1 of referee 1. The title suggested by referee 2 is very similar to the title suggested by referee 1, we picked the first suggestion because it's a bit more specific. We will now change the title to "Multi-variable flood damage modeling with limited data using supervised learning approaches".

2. *Second, I have the feeling that some of the existing (and relevant) literature on this topic is not included in the Introduction so far. It would be interesting to see more than the presented references to (mostly) Dutch researchers and the Potsdam group, e.g., by broadening the focus a bit towards works on flooding with sediment transport – here similar problems are described that somehow the deposition height is the only available parameter, but in turn this parameter is not fully representing the processes leading to loss. Examples include the works of Papathoma-Köhle or Fuchs, to just drop some names.*

We thank the referee for the suggestion to widen the scope for the introduction and possibly the discussion. We will mention the vulnerability indicators in Papathoma-Köhle (2016), and Papathoma-Köhle et al. (2014) will be used to very briefly describe the state of the art in landslide vulnerability.

3. *Third, in the discussion on vulnerability of buildings exposed to flood hazards there are some works not comparing direct losses, but the degree of loss, which is a relative measure taking into account the different building values. As such, and I am not com-pletely familiar with Dutch building regulations, different loss heights are also a result of different values of the elements at risk. How did the authors consider this challenge during their analysis (which is also perfectly mirrored by Figure 1)?*

This is an important issue that will be addressed in the revised manuscript. We will mention in the introduction that we aim for predicting absolute damages rather than relative damages. In the discussion we will discuss the advantages/disadvantages of using absolute/relative damages. The values at risk are included indirectly in variables such as living area, footprint area, building year and basement. Making this relative would be useful if exact building values were available. However, since these building values are not available, general rules of thumb would be needed for building values. This would introduce extra errors, and therefore we decided to use absolute flood damages.

4. *Fourth, I kindly would like to suggest that the Results and Discussion (!) sections are more carefully written since so far, the first includes lots of discussion, and the current Conclusion and Discussion section is rather short. This should also include some paragraphs on the uncertainties behind the analysis, as mentioned in the Methods section.*

We agree with the referee. The conclusion and discussion will be revised considerably, as this suggestion was also made by referee 1. The discussion will focus more on the impact of uncertainties in our dataset and the way they might impact the conclusion. The conclusion will focus more on the goodness of fit indicators and the relationship to the limited data. See also our replies to points 7,21, and 23 from referee 1.

5. *Fifth I would like to recommend that the authors show a more detailed situation as the one presented in Figure 2 – the current scale is hardly readable. A possible solution is to show the overall extent as an inlet map and then in the main map just a zoom of the most interesting river section or so. For the legend: the water depths of 0.5, 1.0 and 2.0 m are not clearly distinguishable, and technically should be presented differently (e.g., by using the ">"). For some of the other Figs. presented I also would like to recommend to clearly state the abbreviations (e.g., td, sd, cd,: : :) in the Figure caption.*

We will add extra zoomed in maps of an interesting river section. We will also change the legend colors and improve the notation in the legend. In case of abbreviations in the figures, we will add the meaning to the figure caption. In cases of figures with many abbreviations, we will reference to the table that has the meanings listed.

6. *Finally, I would recommend to extent the discussion on Fig. 5 – as already indicated there may be variables other than the water height responsible for the loss height available…*

The discussion of figure 5 will be extended and moved to the discussion section (see also our replies to the comments of the first referee).

**References (not already included in the manuscript):**

Papathoma-Köhle, M., 2016. Vulnerability curves vs. vulnerability indicators: application of an indicator-based methodology for debris-flow hazards. Nat. Hazards Earth Syst. Sci., 16, 1771–1790, 2016

Papathoma-Köhle, M., Zischg, A., Fuchs, S., Glade, T., Keiler, M., 2014. Loss estimation for landslides in mountain areas – An integrated toolbox for vulnerability assessment and damage documentation. Environmental Modelling & Software 63 (2015) 156-169.

Penning-Rowsell, E.C., C. Johnson, en S. Tunstall. The benefits of Flood and Coastal Risk Management: A Manual of Assessment Techniques. Middlesex University Press, London, 2005.

Scawthorn, C., et al. „HAZUS-MH flood loss estimation methodology II. Damage and loss assessment." Natural Hazards Review, Vol. 7, No. 2, 2006

Sluijs, L., M. Snuverink, K. van den Berg, en A. Wiertz. Schadecurves industrie ten gevolge van overstromingen. Tebodin. Rijkswaterstaat DWW., 2000.

Thieken, A.H., A. Olschewski, H. Kreibich, S. Kobsch, en B Merz. „Development and evaluation of FLEMOps - A new flood loss esimation model for the private sector." WIT Transactions on Ecology and the Environment 118 (2008): 315 -324.

---

## Author Comment (AC2) · 21 Apr 2017

Thank you very much for your thoughtful and interesting comments. They contribute a lot to further improving our manuscript better. Attached is a pdf with our detailed response to each comment, note that is also includes the comments of referee 1.

Please also note the supplement to this comment:
http://www.nat-hazards-earth-syst-sci-discuss.net/nhess-2017-7/nhess-2017-7-AC2-supplement.pdf
* * *

---

## Author Response (AR2)

[revised manuscript text omitted]

**Reviewer 1**

The efforts the authors have put into the revision of the manuscript have improved the quality of the paper. However, I still feel that the paper would need further rework, especially as far as the methodological sections are concerned.
We thank the reviewer for thoroughly reading our paper again, and for the useful suggestions that are made. Below, we address these one by one.

Have all variables been used as-is in the multiple linear regression model? I am wondering if transformations of certain variables might to better results in the multiple linear regression model?
Yes all variables were used as is. Transformations of input variables could potentially improve the linear regression model. However, in our case we see no indication that transformations would be more suitable than linear regression without transformations. For example, figure 1 doesn't clearly indicate some non-linear relationship in the data. Trying different transformations and testing for the best performance is possible but would mean adding more models, which is outside the scope of the paper since we already have a large number of models. In the linear regression description we added something about transformations.

Lasso does not only perform penalization/shrinkage, but also variable selection. Variable selection is not mentioned in this context, even though it probably is important in case of the multiple linear regression model as well. I find it somewhat difficult to believe that not a single variable is excluded by the Lasso, given that there are certain variables with correlation coefficients to the target variable of as low as 0.01 (Table 1).
LASSO isn't used in the end because the model is insensitive the alpha variable and basically nothing scores better than an alpha of zero. LASSO with an alpha of zero is equal to the ordinary least squares method and so LASSO isn't used here. Some variables have however very small coefficients but aren't completely zero until an alpha of 10. We added this to the variable importance section and compared these variables to the variables from the variable importance from the bagging tree method.

Alpha values for Lasso are usually derived by means of cross validation. It is unclear why and how 4 specific alpha values were selected in this study. Lasso Trace Plots would be a nice graphical illustration that could be added to the manuscript. In addition, it is not clear how the "best" fitting lasso model was selected out of these 4 models.
While alpha value optimization is important in some cases, it is not relevant here, since our model is largely insensitive to the alpha. The discussion section already includes a paragraph about using cross-validation instead of validation on a single test set, we expanded that point to also include the alpha value selection.

Actually, scikit-learn does not support post-pruning, but pre-pruning is available (e.g. specifying depth of tree or split criteria, as implemented by the authors).
That is correct and we used pre-pruning as you note here and we described the process in detail only without mentioning the term pre-pruning. The term is not very commonly used and I think it might be confusing since the term pruning usually refers only to post-pruning.

It is not clear to the user how pruning has been implemented. How does this work, apart from plugging the data into some Matlab toolbox? Has the tree size been selected according to the minimal CV error, or did the authors apply other methods to determine the tree size?

What we did was we build the entire tree until no more splits could be made, then we scaled it back to

the level so that it performed best on the test set. We found the best pruning level based on the MAE. We now added to the revised paper that we used the MAE for finding the best pruning level.

Conceptual confusion is created by somewhat randomly using metrics like (R)MSE, MBE and MAE. For instance, the authors state a 20% reduction in the MAE in the abstract, while they state a 30\% reduction in the MSE in section 3.3.

We understand the MSE might be a bit confusing. This metric is only used within the tree based methods to build the tree, it is not used to evaluate the resulting trees on the test set. The variables all have a very specific purpose, the first paragraph of section 3.1 describes this in detail. An MAE error is the expected error on an individual object, the MBE is the bias error on the aggregate damages (so overestimates are allowed to counter underestimates). The 30% use of variables outside the original dataset cannot be interpreted as an improvement between models, it can only be used relatively among variables. We clarify this now in sections 2.3 and 3.3.

I do not understand why has the RMSE been replaced with the MAE in section 3 and in the tables? The RMSE gives more weight to large errors than the MAE, thus being a nice metric if large errors are particularly undesirable. Why has this been dropped, instead of keeping the RMSE and simply adding the MAE (or rather, adding the RMSE, since the MAE has seemingly been only labelled incorrectly)?

We agree that we should have added some explanation about this to our earlier response. We found out in our code that the error metric we were calculating was the MAE and not the RMSE. We therefore decided not to change the results but to correctly label the error metric instead.

Section 3.3: The total importance of variables that were added in this study is about 30\%. I assume this does refer to the test set? Please clarify.

Actually no, variable importance is calculated during the training. It is therefore based on the training set. As noted before this is really a different measure from the other error metrics, only to be used to see which variables are used during the training of the tree based models. So the 30% improvement is really only for variable importance. We clarify this now in section 2.3 and 3.3.

I would advise to refrain from justifying decisions based on the software used. While it is of course essential to reference the software used, providing explanations related to the methodology itself rather than based on some function arguments is way more proficient:
Thank you for the advice. We agree that we sometimes mentioned the software library when there wasn't really a need for it.We changed this now, see below for more details.

This library requires an alpha parameter to be set (...)} -- actually, this is a core feature of Lasso rather than an attribute of the Python library.
We replaced the "This library" with "The LASSO method"

This was done using Matlab's Statistics and Machine Learning Toolbox (...)} Ok, but how does this work? Which criteria have been used for pruning, i.e. how did the authors derive the optimally pruned tree? There is a reference to Breiman et al., but adding this essential information (1-2 sentences) to a manuscript targeted at a comparison of methods is important to the interested reader. We had about 2 lines introducing the basic idea behind pruning (just before we mention the library applied). Also, we now added some extra details to this introduction, as this is important information indeed.

Section 2.3: It is sufficient to say that the variable importance has been calculated as the normalized total reduction of the MSE, there is no need to reference scikit-learn twice and simply base your decision on the fact that this is feature of scikit-learn. Agree, we removed that sentence.

Figure 5: I would strongly advise to refrain from using pie charts, especially in the way they are presented here, since they are very difficult to read. There are too many variables displayed in random order, using very similar colors. Grouped bar charts or similar plot types would be preferable. We changed to bar charts now. However, the problem with the similar colours remains. The colours are in the order of the legend now though, and the main message of the plot is the importance of water depth compared to the other variables. That is still clear despite the colours.

Figure 6: Please show the number of observations in the plot, e.g. above each box between y-values of 60000 and 70000. In addition, I am wondering why the construction of boxplots differs from the commonly used way of using the first and third quartile as box (plus a band inside the box indicating the median, which is also missing in this case). By the way, using the 5th and 95th percentile as whisker ends is not very common as well, but since it is indicated in the caption it is okay in this case.

We had no reason not to follow the conventions. Therefore, we changed the plot now to follow the conventions. We only kept the 5th - 95th percentile range of the whiskers because our dataset has large random outliers. Also we placed the number of samples above the chart rather than in the caption.

**Reviewer 2**

Dear colleagues,

the revised version of your manuscript increased in clarity and will be a good contribution to the topic. Technical correction: check on page 1, line 32 the reference to Papathoma-Köhle; it should read as "Papathoma-Köhle et al., 2015". The corresponding reference on page 18, line 11: delete "2014." after the author names, paper has been published in 2015.
Further technical corrections: the manuscript style for References has not been used throughout. Should be corrected before final submission (authors may want to download the EndNote Style File at the webpage of NHESS).
We changed the references according to the journal standards and corrected the year of publication.